# The impact of Secondary Ice Production on Arctic Stratocumulus

Georgia Sotiropoulou[1], Sylvia Sullivan[2], Julien Savre[3], Gary Lloyd[4], Thomas Lachlan-Cope[5], Annica M. L. Ekman[6], Athanasios Nenes[1,7]

[1]Laboratory of Atmospheric Processes and Their Impacts, School of Architecture, Civil & Environmental Engineering, École Polytechnique Fédérale de Lausanne (EPFL), Lausanne 1015, Switzerland
[2]Department of Earth and Environmental Engineering, Columbia University, New York, 10027, USA
[3]Meteorological Institute, Faculty of Physics, Ludwig-Maximilians-University, Munich, Germany
[4]Centre for Atmospheric Science, University of Manchester, Manchester, M139P, UK
[5]British Antarctic Survey, Cambridge, CB3 0ET, UK
[6]Department of Meteorology & Bolin Center for Climate Research, Stockholm University, Stockholm, 11419, Sweden
[7]Institute of Chemical Engineering Sciences, Foundation for Research and Technology-Hellas, Patras 26504, Greece

*Correspondence to*:  georgia.sotiropoulou@epfl.ch, athanasios.nenes@epfl.ch

**Abstract.** In-situ measurements of Arctic clouds frequently show that ice crystal number concentrations (ICNCs) are much higher than the number of available ice-nucleating particles (INPs), suggesting that Secondary Ice Production (SIP) may be active. Here we use a Lagrangian Parcel Model and a Large Eddy Simulation to investigate the impact of three SIP mechanisms (rime-splintering, break-up from ice-ice collisions and drop-shattering) on a summer Arctic stratocumulus case observed during the Cloud Coupling And Climate Interactions in the Arctic (ACCACIA) campaign. Primary ice alone cannot explain the observed ICNCs and drop-shattering is ineffective in the examined conditions. Only the combination of both rime-splintering (RS) and collisional break-up (BR) can explain the observed ICNCs, since both these mechanisms are weak when activated alone. In contrast to RS, BR is currently not represented in large-scale models; however our results indicate that this may also be a critical ice-multiplication mechanism. In general, a low sensitivity of the ICNCs to the assumed INP, the Cloud Condensation Nuclei (CCN) conditions and also to the choice of BR parameterization is found. Finally, we show that a simplified treatment of SIP, using a LPM constrained by a LES and/or observations, provides a realistic yet computationally efficient way to study SIP effects on clouds. This method can eventually serve as a way to parameterize SIP processes in large-scale models.

## 1. Introduction:

Mixed-phase clouds are a critical component of the Arctic climate system due to their warming effect on the surface radiation balance (Shupe and Intrieri, 2004; Sedlar et al., 2011) and potential impact on the melting of sea ice. These clouds are very frequent in the summer, when they occur about 80–90% of the time and can persist for days to weeks (e.g. Shupe et al., 2011). However, their representation in

mesoscale and large-scale numerical weather prediction and climate models remains elusive (Karlsson

and Svensson, 2013; Barton et al., 2014; Wesslén et al., 2014; Sotiropoulou et al., 2016).

An accurate description of mixed-phase clouds in models requires a solid knowledge of the amount and distribution of both liquid water and ice (e.g., Korolev et al., 2017). Ice crystals and liquid drops form upon preexisting aerosols, termed ice nucleating particles (INP) and cloud condensation nuclei (CCN), respectively. However, the observed ice crystal number concentration (ICNC) can be

orders of magnitude higher than the number of INPs (e.g., Rangno and Hobbs, 2001; Gayet et al., 2009; Schwarzenboeck et al., 2009; Lloyd et al., 2015). The enhanced ICNCs are especially surprising in the high Arctic, which is relatively clean with sparse INPs (Gayet et al., 2009; Morrison et al. 2012). Secondary Ice Processes (SIP) are suggested as the cause to explain this cloud-ice paradox (e.g., Gayet et al., 2009; Lloyd et al., 2015). SIP refers to a variety of collision-based processes that multiply the

concentration of ice crystals in the absence of additional INP (e.g. Field et al., 2017, and references therein). Yet these processes are poorly represented in atmospheric models, resulting in potential errors in the representation of the surface shortwave radiation budget (Young et al., 2019).

The SIP processes known and studied to date include rime-splintering, break-up from ice-ice collisions and drop-shattering. Rime-splintering (RS) is by far the most explored of all SIP mechanisms,

and refers to the production of ice splinters after super-cooled droplets rime onto small graupel (Hallett and Mossop, 1974). This process occurs effectively for temperatures between -3 and -8$^{\circ}$C (Hallett and Mossop, 1974; Heymsfield and Mossop, 1978), when liquid droplets smaller than 13 μm and larger than 25 μm are present (Hallett and Mossop, 1974; Choularton et al., 1980). RS is the only SIP mechanism that has been extensively implemented in weather prediction (e.g. Li et al., 2008; Crawford et al., 2012;

Milbrandt and Morrison, 2016) and climate models (e.g. Storelvmo et al., 2008; Gettelman et al., 2010).

Secondary ice production also occurs from collisions between ice crystals (Vardiman, 1978; Takahashi et al., 1995) that lead to their fracturing and eventual break-up (BR). This mechanism is most effective at colder temperatures than required for RS, around -15∘C (Mignani et al., 2019). There is still little quantitative understanding regarding this mechanism and its dependence on atmospheric and cloud

conditions; whatever is known comes from limited laboratory experimental data (Vardiman, 1978; Takahashi et al., 1995) and small-scale modeling (e.g. Fridlind et al., 2007; Yano and Phillips, 2011; 2016; Phillips et al, 2017a,b; Sullivan et al., 2017; 2018a). Relatively few attempts have been made to incorporate this process in mesoscale models (Hoarau et al., 2018; Sullivan et al., 2018b; Fu et al., 2019).

Recent laboratory studies suggest that ice multiplication at temperatures around -15$^{\circ}$C can also occur from shattering of droplets with diameters between 50 and 100 μm (Leisner et al., 2014; Wildeman et al., 2017; Lauber et al., 2018) with presumably at least one INP that initiates the ice

formation process. Drop-shattering (DS) has been studied with small-scale models (Lawson et al., 2015; Sullivan et al., 2018a; Phillips et al., 2018) and found to be important for a range of atmospheric conditions. Sullivan et al. (2018b) implemented parameterizations for DS and BR mechanisms in the COSMO-ART mesoscale model to study a frontal rainband, which resulted in reduced discrepancies between modeled and observed ICNCs. In contrast, Fu et al. (2019) implemented DS in the Weather and Research Forecasting (WRF) model for simulations of Arctic clouds, but found insignificant ice multiplication.

Nevertheless, the thermodynamic conditions that favor the above mechanisms can frequently occur in the Arctic. In this study, we examine the potential role of SIP during the Cloud Coupling And Climate Interactions in the Arctic (ACCACIA) flight campaign in 2013. Observations of stratocumulus clouds from the summer flights indicate that ICNCs were orders of magnitude higher than the measured aerosol concentrations that can act as INP, suggesting that ice multiplication may have taken place (Lloyd et al., 2015). To investigate this hypothesis, we use a Lagrangian Parcel Model (LPM) that includes SIP descriptions and a Large Eddy Simulation (LES) that provides a realistic representation of the boundary-layer turbulence and thermodynamic conditions.

## 2. ACCACIA

### 2.1 Measurements

The ACCACIA flight campaign took place during March, April and July 2013, in the vicinity of Svalbard, Norway. The main objectives of this campaign were to reduce uncertainties regarding microphysical processes in Arctic clouds and their dependence on aerosol properties. For this purpose, an extensive suite for microphysical and aerosol instruments was deployed (Lloyd et al., 2015; Young et al., 2016). Below, we offer a brief summary of the dataset utilized in this study.

Images of cloud particles collected with a two-dimensional Stereoscopic Probe (2D-S) at 10-μm resolution were used to calculate number concentrations and discriminate particle phase. The measured concentrations were fitted with "antishatter" tips (Korolev et al., 2011, 2013) to mitigate particle shattering on the probe and have further been corrected for shattering effects using inter-arrival time (IAT) post analysis (Crosier et al. 2013). Ice Water Content (IWC) was determined from these data, using the Brown and Francis (1995) mass dimensional relationship: IWC is the sum of the masses of all ice particles recorded by the 2D-S probe, where the mass of each particle is estimated as a function of its diameter.

A DMT Cloud Droplet Probe (CDP) measured the liquid droplet size distribution between 3-50 μm and was used to derive Liquid Water Content (LWC). A GRIMM Portable Aerosol Spectrometer

provided aerosol size distributions within the range 0.25-32 μm. Owing to a lack of direct INP measurements, GRIMM aerosol concentrations with diameter larger than 0.5 μm are used as input to the DeMott et al. (2010) parameterization (hereafter DM) for primary ice nucleation. Basic meteorological measurements (e.g. pressure, temperature, relative humidity with respect to ice) were also provided by Goodrich Rosemount probes.

Previous analyses of ACCACIA observations have shown that ice multiplication, associated with enhanced ICNCs, likely took place in summer, while ice production in springtime mixed-phased clouds was likely driven by primary ice nucleation (Lloyd et al., 2015). For this reason, our study focuses on a summer single-layer stratocumulus case observed on 23 July.

## 2.2 Case study

The data used in this study were collected on July 23, during Flight M194, when the aircraft flew on northerly and southerly headings through a single-layer stratocumulus around 15°E, between 78.2 and 82°N. On this day, a low-pressure system was centered on 85°N 150°W, while high-pressure systems were prevailing in the sampled region, with particularly high pressure over the north of Norway. Flight M194 sampled clouds in the trailing low pressure system. The aircraft sampled mostly downdrafts, ~5 m s$^{-1}$, when flying at ~1 km height and weak updrafts, ~2 m s$^{-1}$, above 2 km. Winds were usually from the west, except the southerly end of the flight track, where south-westerly winds were measured.

In this study, we focus on a single stratocumulus deck observed between 10:00-11:00 UTC, when the aircraft was flying between 80.8-82°N and 14.7-15.3°E (Fig. 1). This case study is chosen as the aircraft flew at relatively low altitudes, providing detailed information about the planetary boundary layer (PBL) structure. During this period a temperature inversion was found between 0.8 km and 1.2 km altitude, about 3°C strong (Fig. 2a). A specific humidity inversion co-existing with the temperature inversion was also observed, with a strength of 0.5 g kg$^{-1}$ (Fig. 2b). CDP measurements further indicate the presence of a stratocumulus layer above the first 0.5 km of the atmosphere, about 450 m deep, with a cloud top residing within the temperature inversion. Such clouds that penetrate the temperature inversion layer are very frequent in the Arctic (Sedlar et al., 2012).

The cloud droplet number concentration ($N_C$) observed within this hour was highly variable, ranging from 0.2 to 68 cm$^{-3}$ (Fig. 2d), while the mean profile peaks at 30 cm$^{-3}$. INP estimates from DM parameterization indicate a maximum concentration of 0.05 L$^{-1}$ measured at -9°C (above the PBL), while the mean INP value is 0.006 L$^{-1}$ for the observed temperature (-10 – 0 °C) and specific humidity (2.5 – 5 g m$^{-3}$) range. However, the mean observed ICNC for the same conditions is 1.43 L$^{-1}$ and 17.8 L$^{-1}$, respectively. The maximum ICNC occurs at T~-5°C, thus much warmer conditions than those that maximum INPs are estimated, suggesting substantial ice multiplication. Considering that the distance

between the cloud base and the surface was more than 0.5 km, while weak to moderate horizontal wind speeds prevailed, about 5.8 m s$^{-1}$ on average in the PBL, ICNC contributions from blowing snow are unlike (Dery and Yau, 1999; Gossart et al, 2017). For this reason we focus only on secondary ice generation from in-cloud microphysical processes.

## 3. Models and Methods

While RS has been extensively implemented in models, BR is more challenging to parameterize as it requires a correct spectral representation of the ice crystals. This representation is more straight-forward in bin microphysics schemes (e.g. Phillips et al. 2017b), but these are computationally expensive, and thus weather forecast and climate models typically incorporate bulk microphysical representations. It is likely that a property-based ice microphysics scheme, like the Predicted Particle Properties (P3) scheme (Morrison and Milbrandt, 2015; Milbrandt and Morrison, 2016) in WRF, can support a more realistic representation of the BR process. This scheme tracks ice mixing ratio, number, mass, and rime fraction rather than number and mass in snow, graupel, and ice crystal categories, as in bulk schemes, whose thresholds can be non-physical. However, in the current version of WRF, P3 considers only two ice categories while at least three are needed for the BR description (see Section 4.2 for a discussion).

For the above reasons, we combine for our investigations a LPM specifically developed for the study of SIP (Sullivan et al., 2017; 2018a) and the MISU/MIT Cloud and Aerosol (MIMICA) LES (Savre et al., 2015), designed for the study of Arctic clouds. The LPM allows for an adequate description of the formation, growth and evolution of cloud droplets and ice particles as they interact with each other, including SIP. The LES provides a three-dimensional description of the cloud system at a high spatial and temporal resolution, which is of similar scale as the observations. The LPM – driven by the LES conditions - is used to quantify the enhancement in ICNCs due to SIP compared to primary ice formation. The ice crystal concentration in the LES (which includes only a description of primary ice) is then enhanced by the LPM result. This coupling between the LES and LPM occurs at every timestep throughout the simulation and consists a convenient way to combine the benefits of a computationally-inexpensive bin model with the high-resolution LES. A detailed description of the modeling components and the overall modeling methods and set-up are described below.

### 3.1 Large Eddy Simulation (LES)

The MIMICA LES (Savre et al., 2015) solves a set of non-hydrostatic prognostic equations for the conservation of momentum, ice-liquid potential temperature and total water mixing ratio with an anelastic approximation. A 4th order central finite-differences formulation determines momentum

advection and a 2nd order flux-limited version of the Lax-Wendroff scheme (Durran, 2010) is employed for scalar advection. Equations are integrated forward in time using a 2[nd] order Leap-Frog method and a modified Asselin filter (Williams, 2010). Sub-grid scale turbulence is parameterized using the Smagorinsky-Lilly eddy-diffusivity closure (Lilly, 1992) and surface fluxes are calculated according to Monin-Obukhov similarity theory.

Cloud microphysics are described using a two-moment approach for cloud droplets, rain and ice particles. Mass mixing ratios and number concentrations are treated prognostically for these three hydrometeor classes, whereas their size distributions are defined by generalized Gamma functions. Cloud/rain drop processes are treated following Seifert and Beheng (2001), while liquid/ice interactions are parameterized following Wang and Chang (1993). A simple parameterization for CCN activation is 185 applied (Khvorostyanov and Curry, 2006), where the number of cloud droplets formed is a function of supersaturation and background aerosol concentration ($N_{CCN}$). Ice nucleation is also parameterized following DeMott et al. (2010). To account for INP loss due to activation, the newly nucleated crystals at each timestep are estimated by taking the INP number ($N_{INP}$) minus the number of existing ICNCs; this is a standard method applied in microphysics schemes that do not treat INPs as a prognostic 190 variable, e.g. Morrison et al. (2005). CCN and INP concentrations are passively advected within the model domain and not depleted through droplet activation or ice nucleation processes. A detailed radiation solver (Fu and Liou, 1992) is coupled to MIMICA to account for cloud radiative properties when calculating the radiative fluxes.

All simulations are performed on a 96×96×128 grid, with constant horizontal spacing d$x$ = d$y$ = 195 62.5 m. The simulated domain is 6×6 km$^2$ horizontally and 1.77 km vertically. At the surface and in the cloud layer the vertical grid spacing is 7.5 m, while between the surface and the cloud base it changes sinusoidally, reaching a maximum spacing of 25 m. The integration time step is variable, calculated continuously to satisfy the Courant-Friedrichs-Lewy criterion for the Leap-Frog method. Lateral boundary conditions are periodic, while a sponge layer in the top 500 m of the domain damps vertically 200 propagating gravity waves spontaneously generated during the simulations. To accelerate the development of turbulent motions, the initial ice-liquid potential temperature profiles are randomly perturbed in the first 20 vertical grid levels with an amplitude not exceeding 0.0003 K.

### 3.2 Lagrangian Parcel Model (LPM)

The ice enhancement from SIP is estimated with an LPM with six hydrometeor classes for small, medium, large ice and liquid hydrometeors (Sullivan et al., 2017; 2018a). Although the bin microphysics is relatively coarsely resolved, it has served as a convenient framework for the study of ice multiplication, and especially the BR process (Yano and Phillips, 2011; Sullivan et al., 2018a).

The six hydrometeor number tendencies are solved with an explicit Runge-Kutta pair for delay differential equations (Bogacki and Shampine, 1989) and coupled to moist thermodynamic equations for pressure, temperature, supersaturation, liquid water and ice mixing ratios, and hydrometeor sizes; the latter are solved with a second-order Rosenbrock solver (Rosenbrock, 1963). CCN activation is represented in the same way as in the LES, while INP concentration is also constrained based on the LES results (see Text S1 and Fig. S1 in Supporting Information). Each resolved hydrometeors type is represented by a characteristic size that is allowed to dynamically vary over time as a function of temperature and supersaturation. Ice hydrometeors are modeled as prolate spheroids to account for their non-sphericity as in Jensen and Harrington (2015).

The characteristic major axis or radius for the LPM bins are 5 μm, 50 μm and 200 μm for the small, medium and large ice particles (e.g. graupels), respectively, and 1 μm, 12 μm, 25 μm for small, medium and large liquid droplets. The number in these classes is denoted $N_i$, $N_g$, $N_G$ and $N_d$, $N_r$, $N_R$ respectively. A typical timescale for ice crystals to grow to medium sizes ($\tau_i$) for convective clouds with updraft velocities $W \sim$ 2-3 m s$^{-1}$ and cloud base temperature $T_{cbh}$ = 0$^{\circ}$C is 7.5 minutes (Sullivan et al., 2017). However, a somewhat longer $\tau_i$ is expected (~9 min) in Arctic stratocumulus conditions with $T_{cbh}$ = -5$^{\circ}$C and $W \sim$ 0.75 m s$^{-1}$ (Sullivan et al., 2017). Although the colder $T_{cbh}$ promotes ice crystal growth, the weaker updrafts have a pronounced opposing effect. Hence for our ACCACIA case, with mean $W \sim$ 0.25 m s$^{-1}$ and mean $T_{cbh} \sim$ -3.5$^{\circ}$C, i.e. weaker vertical motions and warmer temperatures than in the Arctic case in Sullivan et al. (2017), it is reasonable to assume an even slower $\tau_i \sim$ 12.5 min.

The timescale ($\tau_g$) for medium ice particles to grow to large ones can be inferred from the measurements, since the 2D-S instrument can trace ice particles larger than 75 μm. Ice particles with diameters 400 μm or larger are found systematically and at relatively larger concentrations above 830 m (Fig. S2a), hence ~260 m above the cloud base height. The estimated time for a cloud particle with a mean updraft velocity 0.25 m s$^{-1}$ to reach this level, ascending from the cloud base is ~17.5 min. Hence a $\tau_g$=17.5 min is assumed in our LPM simulations, somewhat faster than the timescale adopted in Sullivan et al. (2017).

A similarly empirical determination of the fallout timescale $\tau_G$ of the large ice particles is not possible. For their idealized Arctic simulation, Sullivan et al. (2017) adapted a timescale of $\tau_G$=12.5 min. In our simulations, we tested three timescales: 12.5 min, 17.5 min and 22.5 min. Our results showed no sensitivity to these values. The simulations with $\tau_G$=17.5 min are presented in the main text.

The timescale $\tau_d$ for small droplets to grow to medium ones is set to 5 min, based on Sullivan et al. (2017; 2018a). The timescale $\tau_r$ for medium drops to grow to large ones is constrained based on the LES simulations. The LES produces very few rain droplets with diameters greater 25 μm; the maximum raindrop concentration never exceeds 0.15 cm$^{-3}$ in the LES (Fig. S3a). For consistency, a relatively long

growth timescale is adapted, $\tau_r$=55 sec, which allows for a limited number of droplets to grow to large sizes, comparable to the LES results (Fig. S3b). This set-up is in general agreement with the observation

that very few droplets of diameters > 25 μm were found near cloud top over the ice-pack. The fallout time $\tau_R$ of large rain droplets in the LPM is set to 60 min, the end of the simulated time, as very limited precipitation (generally < 0.1 mm day$^{-1}$) is produced in the LES simulations.

Secondary ice processes in the LPM include: (a) RS, when a medium or large ice particle collides with a large droplet, (b) BR, when a medium ice hydrometeor collides with a large one and (c)

DS, if a raindrop freezes. These processes are included in an ice generation function along with primary ice nucleation (denoted as *NUC* below):

$$G_{ice} = \frac{dN_i}{dt}\bigg|_{NUC} + \frac{dN_i}{dt}\bigg|_{RS} + \frac{dN_i}{dt}\bigg|_{BR} + \frac{dN_i}{dt}\bigg|_{DS}$$

$$= N_{INP} + F_{RS}\left[K_{RS_g}N_g + K_{RS_G}N_G\right] + F_{BR}K_{BR}N_gN_G + F_{DS}K_{DS}N_R$$

where $K_X$ is the gravitational collection kernel and $F_X$ the fragment number generated by process $X$ (where X=RS, BR, DS; in the case of RS – we consider both RS from medium (RS$_g$) and large (RS$_G$) ice particles. Collisional kernels are described as in Sullivan et al. (2017) and are functions of the

relative difference of the terminal velocity of the two colliding particles. Since the ice growth equation for medium and large ice particles has an asymptotic behavior, eventually the sizes for the two bins will converge to the same values and the collisional kernel for these two ice categories will become near-zero. For this reason, a correction in $\Delta u$ has been applied following Reisner et al. (1998), which accounts for underestimation in the rate of collisions when $u_G \approx u_g$:

$$|u_G - u_g| = \sqrt{1.7(u_G - u_g)^2 - 0.3u_Gu_g}$$

This correction is extensively applied in collisions of large particles in popular microphysics scheme, e.g. Morrison et al. (2005). In our model, we apply this only when medium and large ice particle sizes become comparable and the difference between their radius is smaller than 1% of the smaller particle's radius: $r_G$–$r_g$<0.01$r_g$.

The fragment number generated by rime-splintering is formulated on the basis of the laboratory

experiments conducted by Hallet and Mossop (1974), who found a maximum of 360 splinters per milligram of rime generated round -5$^o$C.

$$F_{RS} = 360\rho_w\frac{\pi}{6}(2r_R)^3$$

where $\rho_w$ is the water density and  r$_R$ represents the radius of the large droplet. This process is fully efficient in the temperature range of -4 to -6 $^o$C, while its efficiency is decreased by 50% for temperatures between -8 – -6 $^o$C and  -4 – -2 $^o$C, and set to 5% below the optimal zone (Ferrier, 1994).

The work of Takahashi et al. (1995) is used to describe break-up, assuming that ice

hydrometeors in the medium bin undergo fracturing:

$$F_{BR} = 280(T - 252)^{1.2} e^{-(T-252)/5}$$

However, their experimental set-up was very simplified using cm-size hailballs, while one of the two colliding hydrometeors remained fixed. In our LPM simulations the ice particles in the medium bin grow from 100 μm to mm sizes (not shown), thus using the above formula would certainly lead to overestimation of the number of fragments produced. For this reason, scaling $F_{BR}$ by a factor of 10-100 for size differences is essential (see Section 3.4 for a discussion).

A more physically-based parameterization for BR has been recently developed by Phillips et al. (2017a), which estimates $F_{BR}$ as a function of collisional kinetic energy and depends on the colliding particles' size and rimed fraction. This results in varying treatment of $F_{BR}$ for different ice crystal types and ice habits. Since this parameterization requires several parameters that are not available in our LPM model (e.g ice particle type, habit and rimed fraction), for its implementation a number of assumptions have to be made. First of all, since primary ice particles grow through vapor deposition and move to the second bin, we assume that this bin represents snow. Given the relatively warm temperature range (Pruppacher and Klett, 1997) and after inspection of particle images, planar ice is likely the most representative ice habit of the ACCACIA case. A rimed fraction of 0.4 is also assumed, as lower values do not yield any SIP; $F_{BR}$ becomes less than unity ICNCs are highly underestimated. Finally, the third LPM bin is assumed to consist of sufficiently rimed particles, thus the collision type adapted in our simulation is that of snow-graupel:

$$F_{BR} = \alpha A \left(1 - exp\left\{-\left[\frac{CK_o}{\alpha A}\right]^{\gamma}\right\}\right) \quad (9)$$

$$\text{where}: K_o = \frac{m_g m_G}{m_g + m_G}\left(u_G - u_g\right)^2 ,$$

$$A = 1.58 \cdot 10^7 (1 + 100\Psi^2)\left(1 + \frac{1.33 \cdot 10^{-4}}{D^{1.5}}\right),$$

$$\gamma = 0.5 - 0.25\Psi,$$

$$C = 7.08 \ 10^6 \psi ,$$

$$\psi = 3.5 \cdot 10^{-3} ,$$

$$a = \pi D_s^2$$

where $D_s$ is the equivalent spherical diameter of the smaller ice particle which undergoes fracturing, α is its surface area and $\Psi$ the rimed fraction. $C$ is the asperity-fragility coefficient and $\psi$ is a correction term for the effects of sublimation in field observations by Vardiman et al. (1978). The above description concerns collisions of either planar crystals or snow with $\Psi<0.5$ and diameter 500 μm $< D <$ 5 mm with any ice particle (crystals, snow, graupel or hail) . However, Phillips et al. (2017a) suggest that this parameterization can be used for particle sizes outside the recommended range as long as the input

variables to the scheme are set to the nearest limit of the range.

Drop-shattering is described as function of a freezing probability ($p_{fr}$), parameterized following Paukert et al. (2017), and a shattering probability ($p_{sh}$) based on droplet levitation experiments conducted by Leisner et al. (2014):

$$F_{DS} = 2.5 \cdot 10^{-11}(2r_R)^4 p_{fr} p_{sh}$$

Freezing is allowed only when raindrop size exceeds 100 μm and $p_{sh}$ is a normal distribution centered at -15°C with a standard deviation of 10°C.

The number balance in each class is the generation function at the current time as a source and the generation function at a time delay as the sink, along with aggregation and coalescence processes. Note that aggregation occurs between small and medium ice particles and generates new particles in the largest bin. Similarly, coalescence removes droplets from the small and medium bins and generates new ones in the large raindrop category. A schematic of all these processes is shown in Fig. 3.

Finally, the hydrometeor number tendencies are coupled to the moist thermodynamic equations to account for the changing system supersaturation and thus changes in their size. All LPM equations, except the newly-implemented parameterization by Phillips et al. (2017a), are described in detail in Sullivan et al. (2017, 2018a).

### 3.3 Initial and boundary conditions

The atmospheric profiles used to initialize the LES are based on in-situ observations collected between 10-11 UTC on 23 July (Fig. 2), along the flight track shown in Fig. 1. The fact that the aircraft did not sample vertically through the atmosphere, but flew across a relatively large domain (9 km × 180 km) and over variable surface conditions (Fig. 1), induces some challenges for the design of the control simulation: measurements below the cloud layer and above the temperature inversion (Fig. 2a) are collected over the ocean, whereas the cloud layer is mostly sampled over the marginal-ice zones (MIZ) and the ice-pack. However, the uncertainty arising from utilizing all these measurements to construct the initial vertical profiles (Fig. 2) is not necessarily larger than utilizing reanalysis data at a similarly coarse resolution.

Since our focus is on the cloud layer, we simulate ice-covered surface conditions in the LES. The co-existent temperature and specific humidity inversions, associated with the cloud top height, as observed in Fig. 2, are typical characteristics of the summertime Arctic PBL over sea-ice (Sedlar et al., 2011; Tjernström et al., 2012). Note that cloud characteristics can vary depending on the surface type, i.e. if it is open-water, MIZ or thicker ice: $N_C$ and ICNC are about 40%-45% lower over open-water than over ice during the examined case (not shown), suggesting that optically-thicker clouds persisted over the latter. For this reason we only use cloud measurements collected at latitudes higher than 81.7°N (Fig. 1) and within a 9×33 km² ice-covered area to evaluate the simulated cloud properties.

The wind forcing is set by specifying the geostrophic wind, constant with height, equal to the observed vertical mean value of 5.8 m s$^{-1}$. The surface pressure is set to 1010 hPa, linearly extrapolated from low-level pressure measurements. The surface temperature is set to 0°C and surface moisture to the saturation value, which reflect summer ice conditions. Surface albedo is set to 0.65, representative of the sea-ice melting season (Persson et al., 2002). In MIMICA, subsidence is treated as a linear function of height: $w_{LS} = -D_{LS}z$, where $D_{LS}$ is the large-scale divergence. $D_{LS}$ here is defined through trial and error: to avoid rapid vertical cloud displacements, we prescribe $D_{LS} = 3*10^{-6}$ s$^{-1}$.

A $N_{CCN}$ concentration of 50 cm$^{-3}$ is prescribed, based on measurements of cloud droplet concentrations over the ice-pack (Fig. 2d), while the sensitivity to this choice is further tested (see Section 3.4 and 4.3). Implementing the temperature-dependent DM parameterization in the LES, with mean observed aerosol concentrations (0.6 cm$^{-3}$) as input, results in the development of a purely liquid cloud layer in the LES (see Section 4.3). Given that the uncertainty in the DM parameterization is about one order of magnitude (DeMott et al., 2010), we therefore assume a baseline simulation where INP estimates are multiplied with a factor of five and we further perform sensitivity simulations by increasing this factor (see Section 3.4 and 4.3).

Initial specific humidity and pressure in the LPM are set to the values measured at the cloud base (3.1 g kg$^{-1}$ and 980 hPa, respectively). The LPM is then run over a wide temperature and vertical velocity range to encompass the in-cloud variability encountered during the LES simulation (see Section 3.4). The maximum duration for LPM simulations is set to 60 minutes, but the simulation ends also when the parcel reaches the lowest cloud temperature observed near cloud top, -6.5°C. This condition ensures that parcels do not reach colder temperatures in the LPM than those encountered in the cloud simulated by the LES.

The ice enhancement factors, defined as $N_{ice}/N_{INP}$, where $N_{ice}$ is the sum of ice number concentrations in all 3 bins, are derived from the LPM calculations at the end of the simulation time. These factors are saved in look-up tables and then used by the LES: the concentration of the nucleated ice particles in each LES column is multiplied at each model time-step by an enhancement factor, which is a function of the cloud base temperature ($T_{cbh}$) and the mean cloud updraft velocity ($W$).

### 3.4 Sensitivity experiments

The role of SIP during the ACCACIA case is investigated with the LES, in which the SIP effect is parameterized through look-up tables that encompass the LPM results (see Section 3.3). The LPM is run over a certain range of temperature and vertical velocities, representative of the ACCACIA conditions. These ranges are determined by the 3D fields produced by the LES. Hourly outputs of the 3D LES fields indicate that in-cloud updraft velocities vary between near-zero and ~1.4 m s$^{-1}$ (Figure S4a), while the mean $W$ is ~0.25 m s$^{-1}$ and only 0.2% of simulated $W$ values exceed 0.5 m s$^{-1}$. The

simulated cloud temperatures span from -6.5$^o$C to -1.5$^o$C (Figure S4a); the coldest temperatures are found just below cloud top, while the cloud base temperature varies between -4$^o$C and -2$^o$C. These results are indicative of very weak convection. To cover all LES simulated conditions, the LPM is run for $T_{cbh}$ between -5 and -1$^o$C and vertical velocity, $W$, between 0.25 and 1.25 m s$^{-1}$, with a step value of 0.5$^o$C and 0.25 m s$^{-1}$, respectively, to derive the ice enhancement factors.

The CNTRL simulation corresponds to the LES experiment that accounts for all SIP processes, with BR being parameterized after Phillips et al. (2017a), as this is the only physically-based description available for this process. A simulation with no active SIP mechanism is also carried out, referred as NOSIP in the text. A comparison of these simulations is found in section 4.1.

           To further examine the sensitivity of the CNTRL results to BR formulation, three additional
sensitivity tests are presented in the same section. In these simulations RS and DS are parameterized as in CNTRL, but BR is now based on the Takahashi results scaled by a factor of (a) 10, (b) 50 and (c) 100 (Fig. 4). Considering that Takahashi et al. (1995) used cm-size hailballs for their experiments, case (a) corresponds to mm-size particles undergoing fragmentation, while (b) and (c) to 500-μm and 100-μm, respectively. These LES simulations are referred as (a) SIP_T0.1, (b) SIP_T0.02 and (c) SIP_T0.01,
where the number indicates the magnitude of scaling applied to Takahashi's formula.

           In Section 4.2, the contribution of each SIP mechanism is examined separately. For this purpose the LPM is run with only one mechanism activated at each time and the produced look-up tables are used to conduct additional LES sensitivity tests, referred as RS and BR, to reflect the mechanism that contributes to ice multiplication. DS is found to be completely inactive in the examined thermodynamic
conditions (not shown) and for this reason this process is not further discussed in the text. This behavior is consistent with previous studies that have shown that a relatively warm cloud base temperature is critical for the initiation of DS (Lawson et al., 2017; Sullivan et al., 2018a) and that the Arctic environment does not favor this process (Fu et al., 2019). In addition to the BR simulation, which employs the Phillips parameterization, the more simplified descriptions based on the scaled results by
Takahashi et al. (1995) are also tested; these LES simulations are referred as BR_T0.1, BR_T0.02 and BR_T0.01 to indicate the scaling factor applied.

           In Section 4.3 the sensitivity to the prescribed $N_{CCN}$ concentration is investigated by testing two additional values: 10 cm$^{-3}$ and 100 cm$^{-3}$. This range covers a variety of atmospheric conditions, from very pristine to cases where polluted air has been advected form the south. Note that CCN can be highly
variable in the Arctic, typically spanning the range 10-300 cm$^{-3}$ within the PBL (Jung et al., 2018). Two different set-ups are used for these tests: (a) similar to the CNTRL simulation with all SIP mechanisms activated, including Phillips parameterization for BR, and (b) no active SIP mechanism. These LES simulations are referred to as (a) CCN10 and CCN100, and (b) CCN10_NOSIP and

CCN100_NOSIP, respectively.

Finally, in section 4.4 the sensitivity to primary ice nucleation is examined. The standard DM parameterization predicts concentrations $<\sim 0.03$ L$^{-1}$ for temperatures $<\sim 6.5^{\circ}$C, which is very close to the upper limit of INP measurements in the Arctic for the given temperature range (Wex et al., 2019). However, when applied in the LES, it does not produce any cloud ice (see section 4.1). For this reason all LES simulations in sections 4.1-4.3 are conducted with DM parameterization multiplied with a

factor of 5 (DM×5), while the simulation with the standard parameterization is presented as sensitivity test, referred as DM. DM×5 predicts INP concentrations between 0.07 L$^{-1}$ at cloud base and 0.11 L$^{-1}$ at cloud top, which is still reasonable for Arctic conditions (Wex et al., 2019). Considering however that the uncertainty in this ice nucleation scheme is a factor of 10 (DeMott et al., 2010), an additional test DM×10 is also performed; the maximum INP concentration near cloud top predicted by this simulation

is 0.3 L$^{-1}$, which is likely an overestimation for Arctic clouds (Wex et al., 2019). Finally an extreme case, DM×100, is also tested where the predicted INPs are now of the same order as the ICNCs observed during ACCACIA.  The simulations that have the same set-up as CNTRL but a different ice nucleation scheme are referred as DM, DM10 and DM100 in the text, while those that do not account for SIP are DM_NOSIP, DM10_NOSIP and DM100_NOSIP.

A summary of all LES experiments is offered in Table 1. All simulations are run for 8 hours; the first 4 hours are considered as spin-up period.

## 4   Results

### 4.1   The impact of SIP on cloud macrophysics and structure

The influence of SIP on Arctic stratocumulus is quantified by comparing the CNTRL and NOSIP LES

simulations with ACCACIA measurements (Fig. 5). The ICNCs ($N_{ice}$) produced by the CNTRL simulation fluctuate between 1.2-1.5 L$^{-1}$, which is in good agreement with the median observed values, but somewhat underestimated compared to the mean. The modeled mean profile of mass mixing ratio ($Q_{ice}$) is also close to the median observed profile, but somewhat lower compared to the mean. In contrast, only including primary ice formation produces ICNCs below the observed range (Fig. 5a),

while $Q_{ice}$ profiles agree with only the lowest values observed (Fig. 5b).

The sensitivity of our results to the newly-implemented Phillips parameterization for BR is also examined in the same figure, by comparing CNTRL to LES simulations that employ the Takahashi scheme.  The mean $N_{ice}$ values in SIP_T0.1 are larger than the median and mean observations, however the modeled ICNCs can explain some of the largest values observed. SIP_T0.02 produces mean $N_{ice}$ and

$Q_{ice}$ profiles in very good agreement with the mean observations, while SIP_T0.01 performs similarly to CNTRL. The differences/similarities between these LES experiments are also reflected in the LPM

results (Text S2, Fig. S5): for the dominant thermodynamic conditions ($W<\sim0.5$ m s$^{-1}$ and -4$^o$C<$T_{cbh}$<-2$^o$C) Phillips parameterization (CNTRL) and SIP_T0.01 predicts an enhancement factor of ~20, while SIP_T0.1 and SIP_T0.02 produce a maximum enhancement of 1.5 and 2 orders of magnitudes, respectively.

An interesting finding is that all simulations that account for SIP produce ICNCs within the observed range, while NOSIP clearly underestimates observations. These results indicate that SIP can indeed explain the observed concentrations, despite the uncertainties in BR parameterization. The SIP_T0.02 simulation, which is in good agreement with mean observations, represents fragmentation of 500-mm particles, while SIP_T0.01 is more representative of 100-mm sizes. Phillips parameterization accounts for different sizes, however it is constrained for a specific collision type and specific particle properties (habit, rimed fraction, etc.). Nevertheless, in reality more than one collision type can happen simultaneously, while the habit and rimed fraction of the particles that undergo fracturing can vary. Moreover, in our LPM model each bin category is represented by a single diameter, while observations indicate a broad particle size spectra, up to 1.27 mm (Fig. S2b). Thus in reality μm and mm particles can undergo break-up simultaneously, which might explain the wide range of observed ICNCs in Fig. 5a.

**4.2 The role of the underlying SIP mechanisms**

To quantify the contribution of each SIP mechanism, simulations that account for a single SIP mechanism are compared in Fig. 6. RS produces mean $N_{ice}$ and $Q_{ice}$ profiles that can explain only the lowest range of the observed concentrations. BR produces somewhat lower concentrations and mixing ratios than RS, and so does BR_T0.01, since this parameterization predicts similar enhancement factors as Phillips et al. (2017a) when implemented in the LPM (see Fig. S6). BR_T0.02 has a more pronounced multiplication effect than RS, however it still underestimates the mean and median observed profiles. BR_T0.1 is the only simulation that results in similar mean cloud properties to the observed.

The weak multiplication effect in RS, BR and B0.01 is also clearly manifested in the LPM results (Text S2, Fig. S6), which in weak updraft conditions produce enhancement factors $<\sim5$, while BR_T0.02 produces up to a 10-fold enhancement. The multiplication factor in BR_T0.1 can vary between 10-100 times for ACCACIA conditions, resulting in improved LES results (Fig. 6) compared to the previous set-ups. However, in this simulation the results of Takahashi et al. (1995) are scaled assuming mm-size particles, which is rather an upper limit for the ice particle sizes measured during the campaign (Fig. S2b).

Figures 4-5 indicate a strong ice generation feedback between RS and BR, which results in

substantially enhanced multiplication compared to the effect that each mechanism can have when acting alone. The new fragments ejected during rime-splintering contribute to more ice-ice collisions and thus further feed the BR multiplication process, which eventually becomes more efficient than RS (not shown). Since BR is parameterized assuming mm-size particles in BR_T0.1, which is the upper bound in observations (Fig. S2b), we suggest that the observed ICNCs are most likely caused by a combination of both mechanisms (Fig. 5a).

While RS has been extensively implemented in mesoscale and climate models, this is not the case with BR; however, our results indicate that this is also an important SIP mechanism. Our findings are in contrast to the results of Fu et al. (2019), who found that BR efficiency is limited in mesoscale simulations of autumnal Arctic clouds. However, apart from focusing on different thermodynamic conditions, another difference is that they performed offline calculations of the BR effect using the parameterization of Vardiman (1978). Another interesting fact is that while other studies (Yano and Phillips, 2011; 2016) have shown that BR can be highly effective at very cold temperatures ($\sim -15^{o}$C), resulting even in explosive multiplication, in the examined conditions it acts as a weaker source of secondary ice, which in combination with RS can still significantly modulate the microphysical state of the cloud.

Following the formula of Yano and Phillips (2011), we estimate the BR multiplication efficiency $\hat{C} = 4C_o \tilde{a} \tau_g \tau_G$, where $C_o$ is the nucleation rate applied in the LPM and $\tilde{a}= \alpha F_{BR}$; $\alpha$ is the sweep-out rate (adapted from Yano and Phillips, 2011). Phillips et al. (2017a) and Takahashi et al. (1995) parameterization, scaled with a factor 50-100, predict $<\sim 5$ fragments per collision in the temperature range of interest (Fig. 4), thus using the upper limit $F_{BR} = 5$ in our calculations yields $\hat{C} = 10.58$, which is similar to the value $\hat{C} = 10$ predicted in Phillips et al. (2017b). Thus the theory predicts an increase of the cloud ice concentration by a factor of ~10 over a time scale of about an hour; we assume that this is likely the maximum efficiency of BR process in Arctic stratocumulus, since 60 minutes is an upper cloud mixing timescale for such clouds.

### 4.3 Sensitivity to CCN concentration

In this section, we examine the sensitivity of our results to the prescribed CCN concentration. The LES is run for two additional $N_{CCN}$ conditions: 10 and 100 cm$^{-3}$ (Fig. 7). The look-up tables used to parameterize SIP in these simulations are shown in Fig. S7.

Distinct differences are observed in cloud droplet concentrations in Fig. 7a, which are significantly reduced with decreasing $N_{CCN}$ along with a slight decrease in cloud thickness. There is no clear impact on cloud droplet number concentrations when SIPs are excluded. ICNCs in Fig. 7b are similar for all simulations that do not account for SIP, while no substantial differences are observed in

$Q_{ice}$ profiles (Figs 7c). In contrast, the CNTRL, CCN10 and CCN100 simulations, all produce clearly different results suggesting that increasing CCN concentrations enhances SIP activity. This is mainly due to the increasing efficiency of RS, as more drops are formed to initiate this process (see Text S2, Fig. S7). All simulations accounting for SIP are in better agreement with observations than those with no active SIP mechanism, suggesting that including a SIP parameterization can improve model

performance for a variety of CCN conditions.

## 4.4 Sensitivity to INP concentration

Here we examine the sensitivity of our results to the INP concentration by conducting six additional LES simulations: DM, DM10, DM100, and DM_NOSIP, DM10_NOSIP, DM100_NOSIP (see Table 1

for details). The vertical $N_C$ profiles exhibit no substantial difference between all simulations except DM100, where the cloud appears geometrically thinner (Fig. 8a). This is due to the substantial ice concentration produced in this simulation, which results in glaciation of the lower portion of the cloud (Fig. 8b). Ice properties, however, exhibit distinct differences among all INP sensitivity tests (Fig. 8b, c).

The standard DeMott parametrization (DM) results in ice properties in agreement with the lowest observed values. If no SIP is accounted for (DM_NOSIP), almost no ice is produced (Fig. 8b, c). DM10 is in good agreement with the median observations (Fig. 8b, c); however, if SIP is deactivated (DM10_NOSIP), the results agree only with the lowest range of measurements. For extremely high INP conditions, when primary nucleation alone (DM100_NOSIP) can produce the mean observed ICNCs,

activating SIP results in mean concentrations of about 4-5 L$^{-1}$, while the simulated mean $Q_{ice}$ profile is close to the observed mean.

          The comparison of $N_{ice}$ profiles between DM and DM_NOSIP simulations suggests that the enhancement due to SIP is about a factor of 50-100, while for CNTRL and NOSIP (DM×5) it is a factor of 15-20 (Fig. 8b). For DM10 and DM10_NOSIP the enhancement is also about one order of

magnitude, while somewhat smaller when comparing DM100 and DM100_NOSIP (Fig. 8b). Thus in the LES simulations, SIP enhancement decreases with increasing primary nucleation. In contrast to the LES, the LPM results suggest that increasing INP concentrations result in more effective SIP (Fig. S7); this result is somewhat expected since larger concentrations of primary ice crystals would result in more frequent ice-ice collisions (Text S2, Fig. S8). However, the LES simulations indicate that processes that

act as sinks for crystal concentrations, such as precipitation, become more effective with increasing $N_{ice}$ and $Q_{ice}$.

          All in all, these sensitivity simulations indicate that considering SIP processes in the LES results in an overall better representation of the cloud ice properties for a variety of INP conditions. Note that

the uncertainty in the DM parameterization is about a factor of 10 (DeMott et al., 2010) and simulations that predict primary ice within this uncertainty range are in better agreement with the observations when SIP is active. It is interesting to note that even the unrealistic case of DM100 still produces results within the observed $N_{ice}$ and $Q_{ice}$ range, suggesting that a SIP parameterization does not degrade model performance even when unrealistically high INP conditions are prescribed.

## 5. Discussion and conclusions

Semi-idealized simulations of Arctic stratocumulus clouds observed during the ACCACIA campaign are performed to investigate the impact of SIP using a LES and a LPM: the LES provides a realistic representation of the atmospheric thermodynamics, while the LPM provides a more simplified framework to parameterize SIP. The effect of three SIP mechanisms, rime-splintering (RS), collisional break-up (BR) and drop-shattering (DS), is investigated. Furthermore, the sensitivity to the choice of the BR description is also examined, using ice fragmentation rates from Phillips et al. (2017a) and Takahashi et al. (1995); the first parameterization is more advanced, accounting for changes in collisional kinetic energy of the colliding particles, while the latter is a more simplified temperature-dependent relationship. Our simulations indicate that SIP processes are essential to reproduce the observed ICNCs, which are well above the concentrations generated by primary nucleation. A good agreement with observed values of cloud ice properties is obtained when either of the BR descriptions is employed, as long as the formula derived from Takahashi et al. (1995) is properly scaled for size and a high rimed fraction is prescribed in Phillips parameterization.

When the contribution of each mechanism is examined separately, DS is found to be ineffective, which is in good agreement with previous studies of Arctic clouds (Fu et al., 2019). Moreover, both RS and BR are weak when being the only active SIP mechanism. The limited influence of RS is due to the lack of relatively large raindrops to initiate this process. RS has also been found insufficient to explain observed ICNCs in Antarctic stratocumulus clouds in a similar temperature range (Young et al., 2019). To reproduce the observations, Young et al. (2019) had to remove the liquid thresholds from the RS parameterization that allow RS activation only when sufficiently large droplets are formed. Furthermore, they had to multiply the RS efficiency by a factor of 10. The limited efficiency of BR is due to a lack of enough primary ice crystals to initiate ice-ice collisions. Our results indicate that the combination of both RS and BR is a possible explanation for the observed ICNCs; the newly generated fragments by RS further fuel the BR process, resulting in substantial ice enhancement through the latter, compared to when only one mechanism is active. Interestingly, when only RS is accounted for, the multiplication effect has to be increased by about a factor of 10-20 to obtain a good agreement with the observed ICNCs, i.e. the same factor as that used in Young et al. (2019).

Our results here indicate that at relatively warm sub-zero temperatures and in low updraft conditions, BR is a potentially important ice production mechanism, particularly in combination in RS. BR efficiency in Arctic conditions has been also documented in observational studies of mixed-phase clouds in the past (Rangno and Hobbs, 2001; Schwarzenboeck et al., 2009). Schwarzenboeck et al. (2009) analyzed measurements collected with a Cloud Particle Imager during the ASTAR (Arctic Study of Aerosols, Clouds and Radiation) campaign and found evidence of stellar-crystal fragmentation in 55% of the samples; 18% of these cases were attributed to natural fragmentation, while for the rest 82% the possibility of artificial fragmentation (e.g. shattering on the probe) could not be excluded. Moreover, they only included stellar-crystals with sizes $>\sim$ 300 μm in their analysis, suggesting that their estimate for natural crystal fragmentation frequency is likely underestimated.

Despite the potential significance of BR, very few attempts have been made to include this process in large-scale models. Hoarau et al. (2018) recently incorporated BR in Meso-NH which includes a two-moment microphysics scheme with three ice hydrometeor types: ice crystal, graupel and snow particles, whose sizes are determined by gamma distributions (as in most bulk schemes). To represent BR, they assumed a constant number of fragments generated when snow collides with graupel. However, this approach may result in significantly underestimated SIP as other type of collisions that include large ice crystals may occur (Phillips et al., 2017a). Sullivan et al. (2018b) did consider collisions between ice crystals and the other two hydrometeor types in a similar bulk scheme in COSMO-ART, using the original (unscaled) formula of Takahashi et al. (1995). However, their approach may instead result in an overestimated BR efficiency, as not all crystal sizes are suitable to fuel this process, including the very small fragments generated by BR. Nevertheless, one of the most important outcomes of this study is that the simple framework of the LPM, when it is driven ("tuned") by the LES thermodynamic fields provides ice number enhancement factors that bridge the model results with observations. This suggests that the LPM, when appropriately constrained by observations (or LES-type simulations), provides a promising approach towards parameterizing SIP in large-scale models.

Our results indicate that BR is likely a critical mechanism in Arctic stratocumulus clouds, where large drops are sparse and RS efficiency is limited. Thus a correct representation of this process in models will likely alleviate some of the model deficiencies in representing cloud ice properties and hence the shortwave radiation budget (Young et al., 2019). However, existing parameterizations are based on old laboratory datasets and simplified experimental set-ups (Vardiman, 1978; Takahashi et al., 1995). As there have been significant advances in the development of laboratory instruments suitable for BR studies through the past decades, we highlight the need for new laboratory experiments with more realistic set-ups that focus on the BR mechanism. We believe that constraining BR accurately in

models could have a significant impact on the representation of Arctic climate in large-scale models and projections for the future.

**Code availability:** The original LPM code can be found on https://github.com/scs2229/SIM. The LES code is available upon request.

**Data availability:** ACCACIA observations are available on https://data.bas.ac.uk and http://www.ceda.ac.uk.

**Author contribution:** GS and AN conceived and lead this study. AMLE and JS provided the LES code, while SS wrote the original LPM code. GL and TLC provided the ACCACIA observations. GS performed the LPM and LES simulations, analyzed the results, and together with AN wrote the main manuscript. AMLE, SS and JS were also involved in the scientific interpretation, discussion, and commented on the paper.

**Competing interests:** The authors declare that they have no conflict of interest.

**Acknowledgements:** We acknowledge support from Laboratory of Atmospheric Processes and Their Impacts at the Ecole Polytechnique Federale de Lausanne, Switzerland (http://lapi.epfl.ch) and the project PyroTRACH (ERC-2016-COG) funded by H2020-EU.1.1. – Excellent Science – European Research Council (ERC), project ID 726165. We are also grateful to ACCACIA scientific crew for the observational datasets used in this study.

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

**Tables:**

**Table 1:** Description of the LES experiments performed in this study

| LES experiment | SIP process active | $N_{CCN}$ concentration (cm$^{-3}$) | INP concentration (L$^{-1}$) |
|---|---|---|---|
| CNTRL | RS, BR (Phillips parameterization) | 50 | DM×5 |
| NOSIP | none | 50 | DM×5 |
| SIP_T0.1 | RS, BR (Takahashi scaled with a factor of 10) | 50 | DM×5 |
| SIP_T0.02 | RS, BR (Takahashi /50) | 50 | DM×5 |
| SIP_T0.01 | RS, BR (Takahashi /100) | 50 | DM×5 |
| BR | BR (Phillips) | 50 | DM×5 |
| BR_T0.1 | BR (Takahashi /10) | 10 | DM×5 |
| BR_T0.02 | BR (Takahashi /50) | 50 | DM×5 |
| BR_T0.01 | BR (Takahashi /100) | 50 | DM×5 |
| RS | RS | 50 | DM×5 |
| CCN10 | RS, BR (Phillips) | 10 | DM×5 |
| CCN10_NOSIP | none | 10 | DM×5 |
| CCN100 | RS, BR (Phillips) | 100 | DM×5 |
| CCN100_NOSIP | none | 100 | DM×5 |
| DM | RS, BR (Phillips) | 50 | DM |
| DM_NOSIP | none | 50 | DM |
| DM10 | RS, BR (Phillips) | 50 | DM×10 |
| DM10_NOSIP | none | 50 | DM×10 |
| DM100 | RS, BR (Phillips) | 50 | DM×100 |
| DM100_NOSIP | none | 50 | DM×100 |

**Figures:**

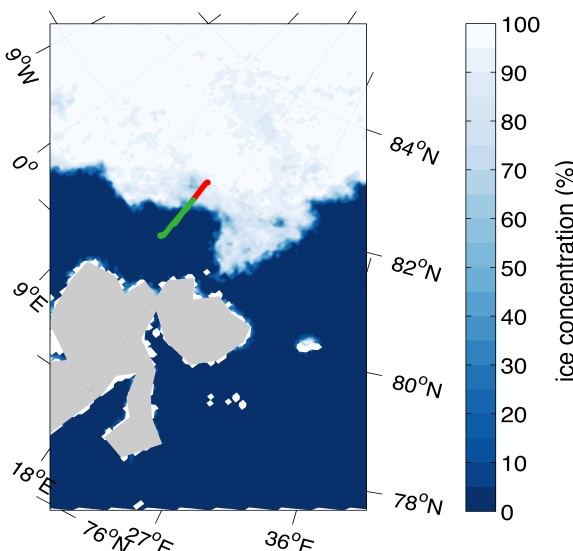

**Figure 1.** Advanced Microwave Scanning Radiometer (AMSR2) daily sea-ice concentrations (grid resolution 6.25 km), from University of Bremen, for 23 July 2013. Green line represents the flight track during ACCACIA campaign, between 10-11 UTC. Red line shows the flight track at latitudes > 81.7°N; measurements collected along this track are used to evaluate the simulated cloud properties.

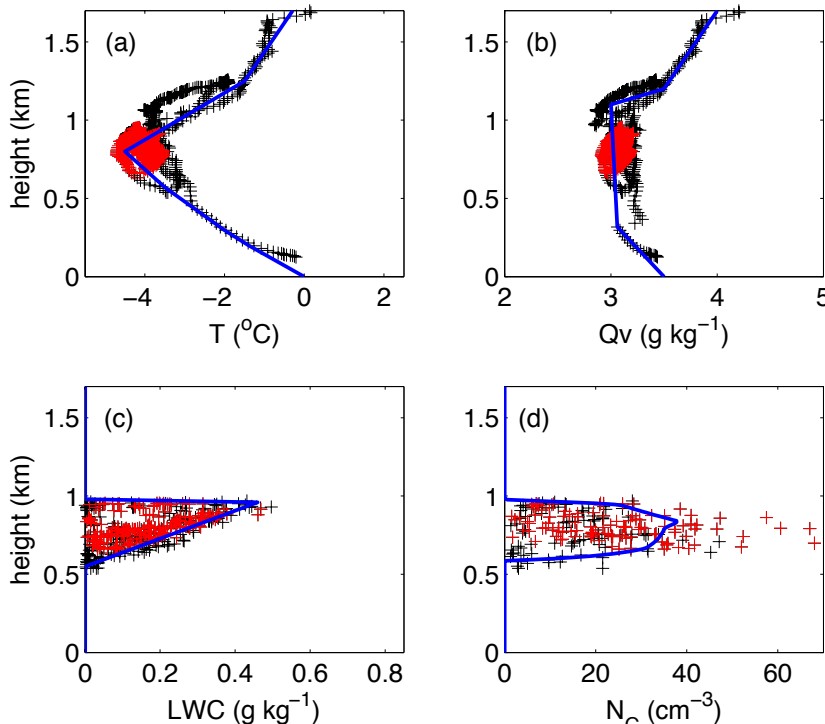

**Figure 2:** Measurements of (a) temperature (°C), (b) specific humidity (g kg[-1]) and (c) liquid water content (g kg[-1]) collected on 23 July 2013 (10-11 UTC) are indicated with black crosses. Red crosses indicate the measurements collected over the ice-pack (above 81.7°N); these are used to evaluate the simulated cloud properties. The blue lines in panels (a-c) represent the simplified vertical profiles used to initialize the LES, while in panel (d) it indicates the cloud droplet concentrations generated by the LES with CCN activation after 1 hour of simulation.

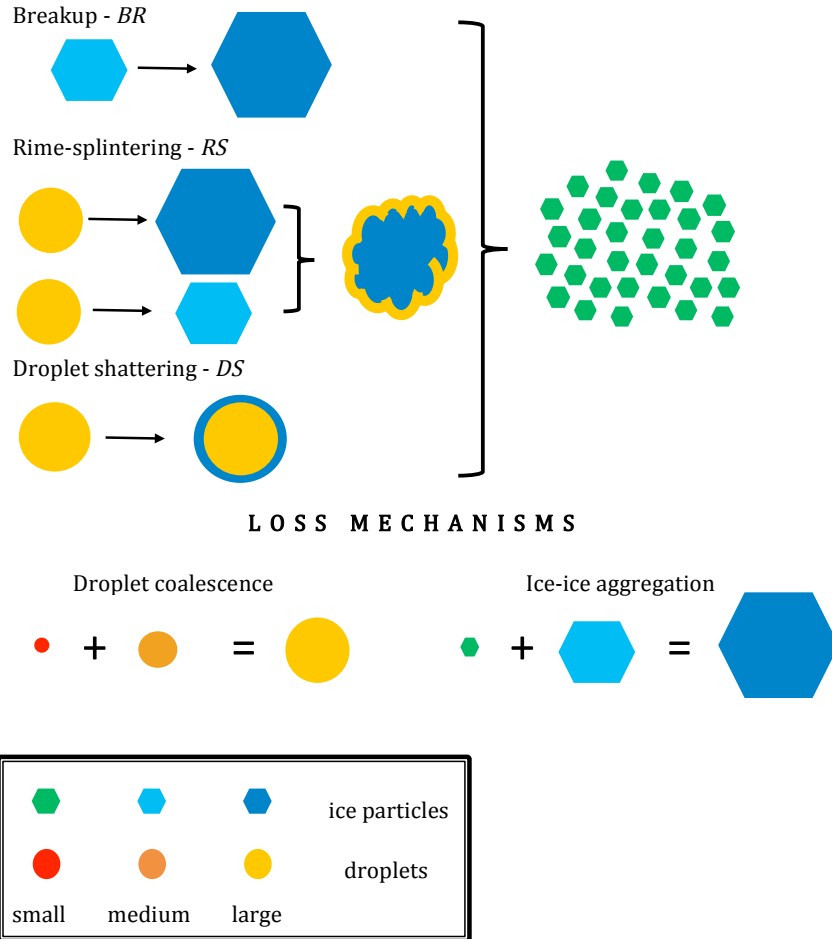

**Figure 3:** Schematic representation of the simplified six-bin microphysics (adopted from Sullivan et al. 2017)

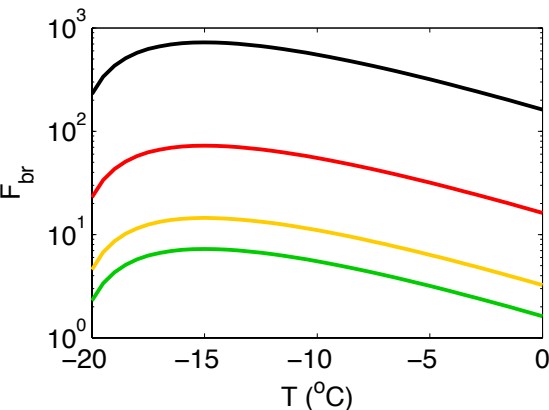

**Figure 4:** Number of fragments generated per collision as a function of temperature estimated with the original Takahashi's formula (black), or scaled with a factor of 10 (red), 50 (yellow) and 100 (green) to represent ice particles of mm, 500-μm, 100-μm size, respectively, that undergo fragmentation.

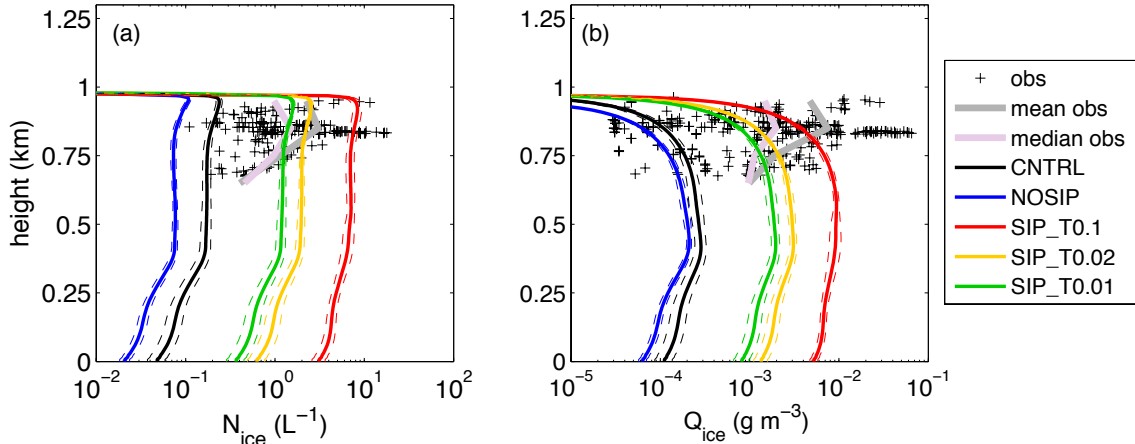

**Figure 5:** Vertical profiles of (a) ice crystal number concentration ($N_{ice}$) and (b) ice mass mixing ratio ($Q_{ice}$) for CNTRL (black), NOSIP (blue), SIP_T0.1 (red), SIP_T0.02 (yellow) and SIP_T0.01 (green) from the LES. Solid lines represent the mean profiles, averaged between 4-8 hours of simulation time, while dashed lines show the standard deviation. Black crosses represent the measurement range derived from the 2D-S Probe, while grey (pink) lines represent the observed mean (median) profiles.

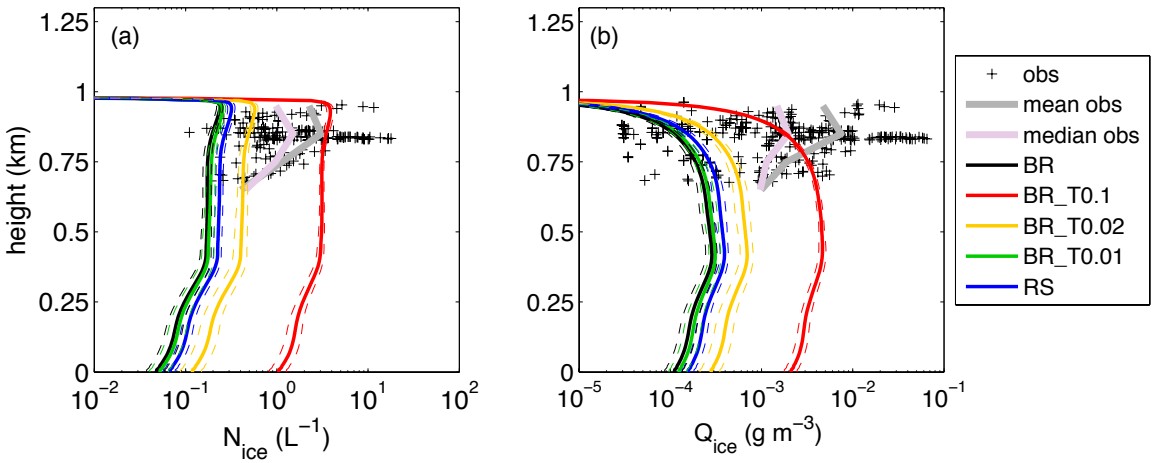

**Figure 6:** Same as in Fig. 5 but for the LES simulations with only one SIP mechanism active: BR (black), BR_T0.1 (red), BR_T0.02 (yellow), BR_T0.01 (green) and RS (blue).

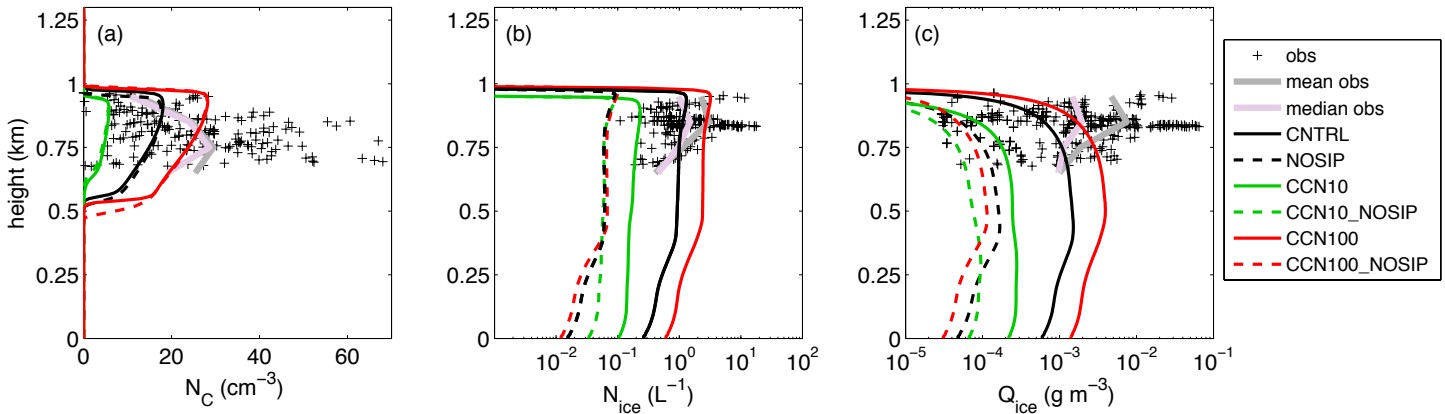

**Figure 7:** Vertical profiles of (a) cloud droplet concentrations (cm$^{-3}$), (b) ice crystal concentrations (L$^{-1}$) and (c) ice mass mixing ratio (g m$^{-3}$) for the LES sensitivity simulations with varying $N_{CCN}$. Black, green and red solid (dashed) lines represent CNTRL (NOSIP), CCN10 (CCN10_NOSIP) and CCN100 (CCN100_NOSIP) runs, respectively. The results are averaged between 4-8 hours of simulation time. Black crosses represent the observations, while the solid grey lines show the median observed profile.

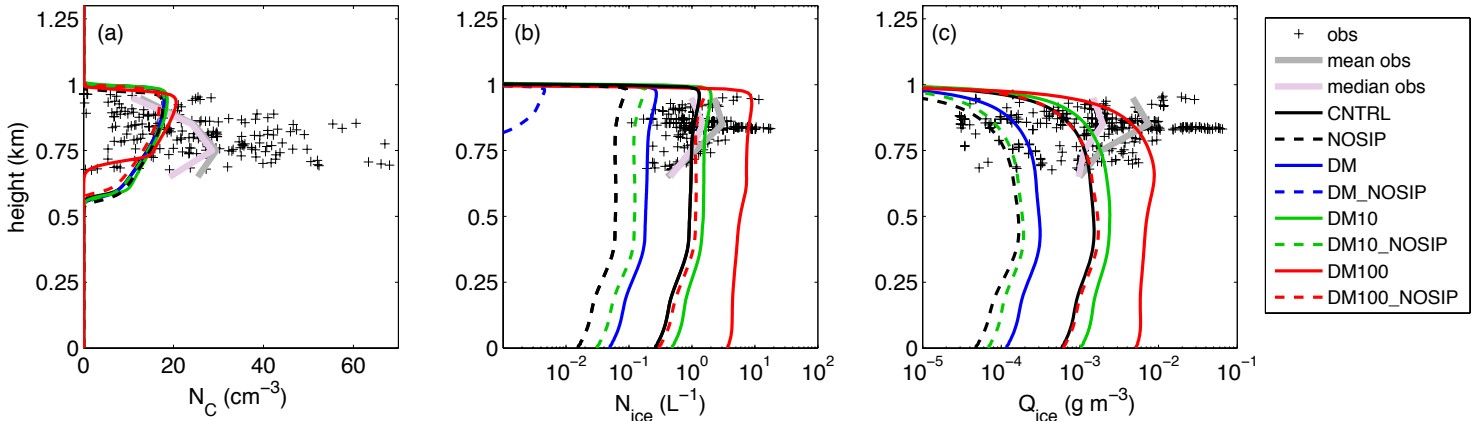

**Figure 8:** Same as Fig. 7 but for the LES sensitivity simulations with varying INP concentration. Black, blue, green and red solid (dashed) lines represent CNTRL (NOSIP), DM (DM_NOSIP), DM10 (DM_NOSIP), DM100 (DM100_NOSIP) experiments, respectively.