# Peer review of "The impact of Secondary Ice Production on Arctic Stratocumulus"

_Atmospheric Chemistry and Physics, 2019_

## Referee Comment (RC1) · Anonymous Referee #1 · 24 Sep 2019

Verdict

I think the paper should be published subject to major revisions.

Major Comments

The structure of the results section could be improved. First, it would be better if there were a comparison of a control simulation with observations (model validation). The control run should be the most realistic of the simulations that the authors can manage, which would be expected to include breakup. Second, once the control run is validated, then the sensitivity tests should be shown, excluding the various processes. Also the description of the simulations with various IN assumptions is vague, with the temperature of each active IN concentration being not mentioned. Also, it is a struggle

to reconcile the model with observations in Fig. 9b. One or two simulations without breakup seem more accurate than those with breakup. Yet in the abstract you write the inclusion of breakup brings the model into agreement with observations for the case.

Detailed Comments

Abstract

The term "droplet-shattering" is used where I think it would be more accurate to say "drop-shattering" or "rain/drizzle-drop shattering". The type of shattering that the authors refer to is of drops > 0.05 mm in diameter, while cloud-droplets are typically smaller than this. Cloud-droplets are < 0.05 mm diameter and are observed not to shatter or splinter.

1. Introduction

Line 168: I thought Savre had developed an ice nucleation scheme with the MISU group. So I wonder why it is not being applied here.

2. ACCACCIA

Page 4, "between 10-11 UTC" should be "between 10:00 and 11:00 UTC".

3. Models and Methods

It is written "Ice nucleation is also parameterized following Morrison et al. (2011): if ICNCs fall below the prescribed INP concentration (NINP), they are nudged upward towards the INP value. " But this seems less accurate than tracking the number concentration of IN lost by activation with a separate prognostic variable, as pioneered by Cohard and Pinty in the 1990s. Computational cost would be minimal. To avoid confusion, it would be a good idea to paraphrase that the "active IN concentration" (I prefer this phrase over "INP" concentration since it is self-evident that the IN is a particle and what is important is the activity spectrum; there is no single number for the concentration) is prescribed from the DeMott 2010 parametrisation informed by total aerosol

measurements of the ACCACIA case.

The DeMott scheme has no dependence on aerosol chemical composition and size. The scheme implicitly assumes that only dust is the IN species, since concentrations in the measurements setting up the DeMott scheme originally involved dust dominating the sizes > 0.5 micron. How does one know that bio-IN were not dominating the IN activity in this case ? Or soot from biomass-burning ? One wonders if another scheme with aerosol chemistry/size dependencies might be more accurate.

A sensitivity test with respect to choice of IN scheme would be a good idea.

There must have been IN measurements in the Arctic in different years, so it would be best to include in the paper a plot of the active IN vs temperature comparing your scheme with the IN measurements from other Arctic campaigns in summertime of various years.

The breakup scheme is based on Takahashi 1995. But they observed collisions between two giant ice spheres (2 cm), one of which was rimed. Phillips et al. (2017a) when building their breakup scheme interpreted these as representing graupel-graupel collisions because the bulk density of the colliding spheres was that of pure ice, not graupel-snow collisions. Can the author comment on this ? Have the authors rescaled the Takahashi data to account for the typical sizes of the graupel in Arctic clouds.

The breakup scheme by Phillips et al. (2017a) is based realistically on collision kinetic energy and temperature, with different treatments for each permutation of species of collisions (graupel-graupel, graupel-snow, snow-snow) etc. It would be better for the authors to upgrade their treatment of breakup.

The authors write "The mean observed INP concentration is 0.006 L-1 and never exceeds 0.05 L-1, while the mean and maximum observed ICNC for the same period is 1.43 L-1 and 17.8 L-1, respectively, suggesting substantial ice multiplication. " But the authors need to say what conditions of temperature and humidity are used to define
these active IN concentrations.

4. Results

The conclusion stated in the Abstract is plausible: "In contrast, break-up enhances ICNCs by 1-1.5 orders of magnitude, bringing simulations in good agreement with observations". However inspection of Fig. 9b comparing predicted and observed ice concentrations shows that the control run without ice multiplication is an order of magnitude too low and with it is an order of magnitude too high.

It seems confusing that the run without ice multiplication is referred to as the "control" and is depicted with a dashed line rather than a full line.

I wonder if the over-prediction of breakup is due to inaccuracy in the formula. Were the Takahashi observations re-scaled for the smaller particles relative to the lab experiment ? Takahashi did when he applied his own lab data to provide estimates for natural clouds.

5. Discussion

The paper by Schwarzenboeck et al. (2009) was seminal and totally relevant as a motivation for the present study. So, there needs to be a more thorough description of their analysis and how they arrived at their estimate of about half (20-80%) of all ice particles being naturally fragmented. They were aware of the shattering bias issue quantified by Field et al. and Korolev et al., and did a diligent study. A few more sentences describing the paper are needed.

Line 449: The comment about the fallout time-scale not being objectively defined could be misinterpreted. What the authors intend to say is that in their own model, the fall-out time-scale can have values in a wide range (there is a similar timescale parameter in the Yano-Phillips theory).

An order of magnitude estimate of the 'multiplication efficiency' (tilde c) for breakup in the model would be helpful, using the formula for it from Yano and Phillips (2011). Although their theory was originally for graupel-graupel collisions, Phillips et al. (2017b) argued it also applies to graupel-snow collisions with a few changes of the parameters. The multiplication efficiency then implies a time-scale for the growth of ice concentration. Does the simulated time-scale of the explosion match the modified theory ?

Does the theory predict that the Arctic clouds simulated is in the unstable regime of the phase-space ?
* * *

---

## Referee Comment (RC2) · Anonymous Referee #2 · 30 Oct 2019

The manuscript studies the sensitivity of ice particle number concentration on different assumptions about secondary ice production in Arctic mixed phase clouds. The topic itself is interesting and merits publication, but selection of methods and the observational case requires more discussion. There is some discussion of the weaknesses of selected modelling approach in Discussion section, but the reason for the employment of certain modelling tools should be made clear already in the model description. Overall, even though the comparison to measured data and selected methods have issues, the manuscript provides some evidence for the role of secondary ice production and gives directions for further studies. Thus I recommend publication after corrections and additional discussion.

1) What is the role of measurement conditions? It is said that the wind is from the West

and measurements are performed both over the open water and ice. Droplet concentration seems similar in both, but there is no discussion how ice particle concentration differs and where the presented values are measured. Is there any potential for the surface to be the source if ice hydrometeors?

2) LPM: I have some problems in understanding what the LPM model employed is actually simulating. Does it solve the hydrometeor condensational growth assuming three size classes both for liquid and ice, or is it somehow parameterized how particles grow from some size range to another based on characteristic time parcel is spending in a single updraft. I'm actually surprised that there is no spectral size resolving model employed. Such models should be available and numerically efficient enough to be used in the presented application. In line 139 it is stated "The LPM allows a detailed description of the formation, growth and evolution of cloud droplets and ice particles as they interact with each other". I disagree with this, I would not call three bins detailed what comes to representation of cloud and ice particle size distributions. Thus also coagulation rate and secondary ice production are only approximate, although probably accurate enough to provide first estimates and to be used in this paper.

3) MIMICA: Line 145 or later in section 3.1: Maybe you should state explicitly the reason why SIP processes are not directly implemented into MIMICA.

4) How does the SIP enhancement work in a case when the ice particle concentration at cloud base in MIMICA is higher than predescriped IN concentration? Does it still enhance the concentration? I assume such conditions to occur frequently in modelled boundary layer cloud.

5) Line 283: "The mean observed INP concentration is 0.006 L-1 and never exceeds 0.05 L-1". From where does these numbers come from? The conditions are really warm for heterogeneous ice nucleation, with modelled values at minimum -6.5 degrees and measured even warmer. What aerosol particles are active in such a warm temperature.

6) Within MIMICA it would be possible to track temperature dependent IN concentration. How would this more realistic approach change the simulations? In comparison to observations it would have been interesting to see if the spread in modelled data is as wide as in observations. When I look at modelled data, I am really surprised how small standard deviation there is in the output. Enhancement should depend quite strongly on the updraft at the cloud base based on Figures 4, 6 and 8.

7) Line 463: "A main challenge in parameterizing BR is that a correct spectral representation of the ice crystals is required, which is more feasible in bin microphysics schemes". This is true, and the same limitations holds for all cases when temperature dependent ice nucleation or secondary ice production is included. If the number concentration is tuned to be correct, the size distribution and total mass is probably wrong due to given shape for size distribution.

8) Jones et al., 2018 is not accepted for publication, so it should not be cited.

9) Schwarzenboeck et al., 2009 title is "Indications for stellar-crystal fragmentation in Arctic clouds"

---

## Author Comment (AC1) · 14 Dec 2019

**RESPONSE TO REVIEWER 1:**

We are grateful for the many insightful and constructive comments, in addition to the suggestions on process descriptions that have clearly strengthened the study. Our responses (in black) to the issues raised (red) are presented below.

**Major Comments**

**The structure of the results section could be improved. First, it would be better if there were a comparison of a control simulation with observations (model validation). The control run should be the most realistic of the simulations that the authors can manage, which would be expected to include breakup. Second, once the control run is validated, then the sensitivity tests should be shown, excluding the various processes.**

Good point. The CNTRL LES simulation is now the one that include all SIP mechanisms and we have attempted to evaluate the validity of the simulations to the fullest extent possible with the observational data available.

**Also the description of the simulations with various IN assumptions is vague, with the temperature of each active IN concentration being not mentioned.**

In the original submission we prescribed a constant INP concentration throughout the domain, which corresponds to the mean primary ice concentration estimated offline with DeMott parameterization (averaged over the observed temperature range). In the revised manuscript we have implemented the aerosol-aware DeMott parameterization in the LES to allow for INP to respond to aerosol concentration and temperature. What is interesting is that despite this adjustment, the LES dynamics tends to mix the INP throughout the cloudy column, so that a quasi-homogeneous INP profile still emerges (see Figure S1 in the revised Supporting Information)

**Also, it is a struggle to reconcile the model with observations in Fig. 9b. One or two simulations without breakup seem more accurate than those with breakup. Yet in the abstract you write the inclusion of breakup brings the model into agreement with observations for the case.**

In Fig 9b in the initial manuscript the only simulation without break-up that gives a better representation of the number concentrations is IN1; this is because an extremely high initial INP concentration $\sim 1$ L$^{-1}$ is prescribed (see Figure 1 below). However, even when we activate SIP in these unrealistic conditions (INP1_SIP) the mean concentration goes up to 2-3 L$^{-1}$, without affecting the ice water mixing ratio. In all other cases (IN0.01_SIP and ALLSIP) activating BR improves the results, as the produced concentrations fall within the observed range. For this reason we state that including BR in the microphysics scheme is likely required to reconcile the simulations with observations. Note that in the revised manuscript, all relevant results are updated with the INP predicted by the DeMott INP parameterization – and the conclusion still remains unchanged.

**Detailed Comments**

**Abstract**

**The term "droplet-shattering" is used where I think it would be more accurate to say "drop shattering" or "rain/drizzle-drop shattering". The type of shattering that the**

**authors refer to is of drops > 0.05 mm in diameter, while cloud-droplets are typically smaller than this. Cloud-droplets are < 0.05 mm diameter and are observed not to shatter or splinter**.

Indeed so. The term is now changed to "drop-shattering"

**1. Introduction**

**Line 168: I thought Savre had developed an ice nucleation scheme with the MISU group. So I wonder why it is not being applied here.**

MIMICA can conduct simulations with the Phillips nucleation schemes (Phillips et al. 2013) and a scheme based on Classical Nucleation Theory (Savre and Ekman 2015). Both however require knowledge of the aerosol composition of the studied atmospheric conditions (number of mineral dust, organic, BC particles), but such measurements are not available during ACCACIA. Nevertheless, to explicitly simulate the heterogeneous ice nucleation process, we have now implemented in MIMICA the DeMott parameterization with mean aerosol measurements from ACCACIA as input.

**2. ACCACCIA**

**Page 4, "between 10-11 UTC" should be "between 10:00 and 11:00 UTC".**

Thank you, corrected.

**3. Models and Methods**

**It is written "Ice nucleation is also parameterized following Morrison et al. (2011): ICNCs fall below the prescribed INP concentration (NINP), they are nudged upward towards the INP value. " But this seems less accurate than tracking the number concentration of IN lost by activation with a separate prognostic variable, as pioneered by Cohard and Pinty in the 1990s. Computational cost would be minimal. To avoid confusion, it would be a good idea to paraphrase that the "active IN concentration" (I prefer this phrase over "INP" concentration since it is self-evident that the IN is a particle and what is important is the activity spectrum; there is no single number for the concentration) is prescribed from the DeMott 2010 parameterisation informed by total aerosol measurements of the ACCACIA case.**

The points raised by the reviewer are well taken. Including INP as a prognostic variable (thus explicitly describe both nucleation and INP recycling processes) is something that requires significant development, and therefore beyond the scope of this manuscript. However, the temperature-dependent aerosol-aware nucleation scheme by DeMott is now directly implemented in the LES. To avoid continuous nucleation with time and excessive production of primary production of ice crystals, we limit activation with the same method as it is done in the standard Morrison scheme in WRF: new nucleated particles = INP (as estimated by DeMott) - existing $N_{ice}$. If this is negative, then no nucleation is assumed to occur. This is a simplified way to account for INPs lost by activation in previous timesteps and can be found in standard microphysics schemes, such as Morrison et al. (2005)

**The DeMott scheme has no dependence on aerosol chemical composition and size. The scheme implicitly assumes that only dust is the IN species, since concentrations in the measurements setting up the DeMott scheme originally involved dust dominating the sizes > 0.5 micron. How does one know that bio-IN were not dominating the IN activity in this case? Or soot from biomass-burning? One wonders if another scheme with aerosol chemistry/size dependencies might be more accurate. A sensitivity test with respect to choice of IN scheme would be a good idea.**

The DeMott scheme is definitely not as advanced as any other aerosol-chemistry-aware scheme.

However, it has been found to perform better in polar conditions than any other temperature-dependent scheme (e.g. Young et al. 2016, Listowski and Lachlan-Cope 2017), so it is considered the best option for cases where aerosol composition information is limited (as in our case). Interestingly enough, when mean aerosol measurements are used as input for -6.5°C (the coldest simulated temperatures) this scheme predicts INP=0.03 $L^{-1}$, which is very close to the upper limit of INPs in Wex et al. (2019) (Figure 1) that include the effect of bioaerosols. Even if the predicted INP concentration of 0.03 $L^{-1}$ is likely realistic (Figure 1), when prescribed in the LES simulations, it does not produce substantial ice. For this reason we have to consider the uncertainty in the DeMott parameterization which is a factor of 10. But multiplying DeMott by a factor of 10 yields very large INP concentrations (~0.3 $L^{-1}$) near cloud top (minimum -6.5°C), which is unrealistic for warm subzero temperatures (Figure 1). For this reason, in our CNTRL simulation we multiply DeMott by a factor of 5, which gives INP concentrations that vary from 0.007 $L^{-1}$ at cloud base (~ -3°C) to 0.11 $L^{-1}$ near cloud top. However, the sensitivity to the assumed INP conditions is shown in the revised text with three additional tests: (a) original DeMott parameterization, (b) DeMott × 10 and (c) DeMott × 100

[Figure]

**Figure 1**: INP measurements conducted by Wex et al. (2019) at four Arctic sites: blue, red, green and yellow shaded areas represent Alert, Utqiagvik, Ny-Alesund and Villum, respectively. Literature data is also included by Petters and Wright (2015) (gray background), Borys (1983, 1989), Bigg (1996), Bigg and Leck (2001), Rogers et al. (2001), Prenni et al. (2007), Mason et al. (2016), Conen et al. (2016), and DeMott et al. (2016). Green and brown symbols represent data from surface-based measurements; black and blue represent airborne measurements. For Rogers et al. (2001), brown indicates data they cited from the literature, with the vertical bar indicating the extent of the reported values.

**There must have been IN measurements in the Arctic in different years, so it would be best to include in the paper a plot of the active IN vs temperature comparing your scheme with the IN measurements from other Arctic campaigns in summertime of various years.**

The most comprehensive Arctic measurements of IN have been recently documented in Wex et el. (2019). The paper includes relatively long-term measurements at four different Arctic sites for all seasons. They also include several Arctic INP datasets from the literature (see Figure 1 above), thus they give a very clear view of the limited concentrations in this region. Since this paper has very recently been published in ACP, we prefer to refer all readers to this very informative paper.

**The breakup scheme is based on Takahashi 1995. But they observed collisions between two giant ice spheres (2 cm), one of which was rimed. Phillips et al. (2017a) when building their breakup scheme interpreted these as representing graupel-graupel collisions because the bulk density of the colliding spheres was that of pure ice, not graupel-snow collisions. Can the author comment on this ? Have the authors rescaled the Takahashi data to account for the typical sizes of the graupel in Arctic clouds.**

Thank you for this suggestion, we had not scaled Takahashi results with size in the initial submission. Considering that Takahashi used cm-size particles, the overestimation in the number of fragments ejected from the collided particle surfaces in our model can vary from one to two orders of magnitude for μm or mm size ice crystals, respectively. For this reason, we conduct a series of sensitivity tests with the LPM in which Takahashi's relationship is reduced by a factor of: (a) 10, (b) 50 and (c) 100. The number of fragments predicted by these parameterizations is given in Figure 4 in the revised manuscript, while the LES results are shown in Figures 5-6. The original formula used in the original submission predicts more than 100 collisions in the temperature range of interest (Figure 4), which is likely a significant overestimation in SIP production.

**The breakup scheme by Phillips et al. (2017a) is based realistically on collision kinetic energy and temperature, with different treatments for each permutation of species of collisions (graupel-graupel, graupel-snow, snow-snow) etc. It would be better for the authors to upgrade their treatment of breakup.**

The point is well taken. Phillips et al. (2017a) requires several parameters that are not directly available by the model, including collision type, ice habit, rimed fraction of the particle that undergoes fragmentation. For each an assumption is made: *i*) as primary ice particles grow through vapor deposition and move to the second bin, we assume that this bin represents snow; *ii*) given the relatively warm temperature range (Pruppacher and Klett, 1997) and after inspection of particle images, planar ice is likely the most representative ice habit of ACCACIA conditions; *iii*) a rimed fraction of 0.4 is assumed, as lower values do not yield any SIP (because the fragments per collision become less than unity) and ice crystal number is highly underestimated. Finally, *iv*) the third LPM bin is assumed to consist of sufficiently rimed particles, thus the collision type adapted in our simulation is that of snow-graupel.

Since Phillips et al. (2017a) is the state-of-the art parameterization, we consider the LES run with this scheme to be the CNTRL simulation, while the more simplified temperature-dependent parameterizations are presented as sensitivity tests.

Finally, at this point we would like to highlight another modification in our LPM set-up in the revised manuscript. In the initial submission, LPM simulations were run either for 30 minutes or until the LPM temperature reaches -6.5$^{o}$C which is the minimum cloud-top temperature simulated by the LES. The cloud mixing timescale $\tau_{mix}$ was set to 30 minutes, considering

$$\tau_{mix} = \frac{cloud\ depth}{mean\ updraft\ velocity}$$

However inspection of the LPM simulations with a mean updraft velocity of 0.25m s$^{-1}$ and cloud-base temperature = -3.5$^{o}$C revealed that cloud temperature drops to -5.5$^{o}$C within 30 minutes; to reach the minimum LES temperature, another ~14 minutes of simulation are required. Hence, we run all LPM simulations for an hour instead, and let the cloud-top temperature threshold determine the actual length of each simulation.

**The authors write "The mean observed INP concentration is 0.006 L-1 and never exceeds**

**0.05 L-1, while the mean and maximum observed ICNC for the same period is 1.43 L-1 and 17.8 L-1, respectively, suggesting substantial ice multiplication. " But the authors need to say what conditions of temperature and humidity are used to define these active IN concentrations.**

These statistics are based on all the flight data collected on 23 July 2013,between 10:00 and 11:00 UTC, at a range of latitudes, longtitudes and altitudes. They aim to provide a more general overview and also allow for comparison with other ACCACIA flights (e.g. Lloyd et al. 2015) in which INP conditions are estimated and presented in a similar way. In retrospect, we see that this discussion may be too vague, and is now clarified: $0.006L^{-1}$ is the mean concentration is for the whole flight, which sampled at temperatures between $\sim -10\,^{o}C -0\,^{o}C$ and specific humidity $\sim 2.5-5$ g m$^{-3}$. The maximum INP concentration is observed at $\sim T= -10\,^{o}C$ and Qv=2.5 g m$^{-3}$. The maximum ICNC occurs at $T\sim-5\,^{o}C$, much warmer conditions than those that maximum INPs are measured, suggesting that SIP may be occurring.

**4. Results**
**The conclusion stated in the Abstract is plausible: "In contrast, break-up enhances ICNCs by 1-1.5 orders of magnitude, bringing simulations in good agreement with observations". However inspection of Fig. 9b comparing predicted and observed ice concentrations shows that the control run without ice multiplication is an order of magnitude too low and with it is an order of magnitude too high.**

We have now addressed this in the revised manuscript (after scaling Takahashi's results and also testing Phillip's parameterization; see Figure 5 in the revised version).

**It seems confusing that the run without ice multiplication is referred to as the "control" and is depicted with a dashed line rather than a full line.**

Sorry for that, in the revised version CNTRL simulation is always represented with a full line.

**I wonder if the over-prediction of breakup is due to inaccuracy in the formula. Were the Takahashi observations re-scaled for the smaller particles relative to the lab experiment? Takahashi did when he applied his own lab data to provide estimates for natural clouds.**

The revised manuscript now considers three different scaling factors to the Takahashi data; these are presented as sensitivity tests along with the control simulation, which employs Phillips parameterization (Figure 5-6 in the revised manuscript). The scaling factor Fbr/100 is more accurate for particles~ 100µm. The scaling factor Fbr/50 is more representative for particles~500 µm and Fbr/10 corresponds to mm sizes, considering that Takahashi et al. (1995) used cm-sized hailballs in their experiments.

**5. Discussion**

**The paper by Schwarzenboeck et al. (2009) was seminal and totally relevant as a motivation for the present study. So, there needs to be a more thorough description of their analysis and how they arrived at their estimate of about half (20-80%) of all ice particles being naturally fragmented. They were aware of the shattering bias issue quantified by Field et al. and Korolev et al., and did a diligent study. A few more sentences describing the paper are needed.**

The findings of Schwarzenboeck et al. (2009) are now more extensively discussed in the last section of the revised manuscirpt.

**Line 449: The comment about the fallout time-scale not being objectively defined could be**

**misinterpreted. What the authors intend to say is that in their own model, the fall-out time-scale can have values in a wide range (there is a similar timescale parameter in the Yano-Phillips theory).**

This discussion is now removed to avoid any misinterpretation. This paragraph discussed uncertainties in $\tau_g$ which is the timescale for medium ice particles (2[nd] bin) to grow to large graupels. However, your comment suggests that the readers get the impression that this parameter can have values in a wide range, but this is not the case with our simulations. In Yano and Phillips (2011) $\tau_g$ is set 30 min, which was considered an upper limit for deeper convective clouds. In a shallow Arctic stratocumulus layer 30 min can sometimes be the timescale mixing for the whole cloud. Given that ice particles with a diameter ~400 m are found 130 m above cloud base and more systematically 260 m above this level (Figure S2 in the revised Supporting Information) in the observations, the $\tau_g$ in our conditions is shorter than in their study. The adapted timescale 17.5 min is a reasonable approximation, estimated straight from the observations using the mean LES updraft velocity.

**An order of magnitude estimate of the 'multiplication efficiency' (tilde c) for breakup in the model would be helpful, using the formula for it from Yano and Phillips (2011). Al though their theory was originally for graupel-graupel collisions, Phillips et al. (2017b) argued it also applies to graupel-snow collisions with a few changes of the parameters. The multiplication efficiency then implies a time-scale for the growth of ice concentration. Does the simulated time-scale of the explosion match the modified theory?**

$\hat{C} = 4C_o \tilde{a} \tau_g \tau_G$, where $C_o$ is the nucleation rate and $\tilde{a} = \alpha N$, where $\alpha$ is the sweep-out rate and N is the break-up rate. In our case the nucleation rate is estimated about ~ 0.02 $s^{-1}$ $m^{-3}$ , which is calculated as the product of updraft velocity, an assumed lapse rate of 6 K $km^{-1}$, and the temperature derivative of the INP concentration estimated with DeMott × 5 parameterization. Phillips and Takahashi's parameterization scaled with a factor 50-100 predict less than 5 fragments per collision in the temperature range of interest (Figure 4 in the revised version). Thus we use the upper limit N=5 in our calculations of the multiplication efficiency. $\alpha$ is set to 2.4×10$^{-5}$ $m^3$ $s^{-1}$, adapted from Yano and Phillips (2011). Substituting these values in the above equation yields $\hat{C} = 10.58$, which is in excellent agreement with the value $\hat{C} = 10$ cited in Phillips et al. (2017b). This discussion is also added in Section 4.2 in the revised manuscript.

**Does the theory predict that the Arctic clouds simulated is in the unstable regime of the phase-space ?**

This is an excellent question. While $\hat{C} > 1$, which allows the potential explosive multiplication at some point, the limited timescale allowed for SIP to develop is the ultimate limiting factor in Arctic stratocumulus. The theory suggests that over a time scale of an hour, the multiplication is a factor of 10. Given that 60 min is an upper limit for cloud mixing timescale in in these shallow cloud layers, we don't expect ice multiplication to substantially overcome this factor. That said, being in the unstable regime is a requirement for SIP to provide crystals above the primary nucleated concentration.

**References:**

Listowski, C. and Lachlan-Cope, T.: The microphysics of clouds over the Antarctic Peninsula – Part 2: modelling aspects within Polar WRF, Atmos. Chem. Phys., 17, 10195–10221, https://doi.org/10.5194/acp-17-10195-2017, 2017.

Morrison, H., Curry, J.A., & Khvorostyanov, V.I. (2005). A New Double-Moment Microphysics Parameterization for Application in Cloud and Climate Models. Part I: Description, . *Atmos. Sci.,* 62*,* 3683-3704 62

Phillips, V.T., P.J. Demott, C. Andronache, K.A. Pratt, K.A. Prather, R. Subramanian, and C. Twohy, 2013: Improvements to an Empirical Parameterization of Heterogeneous Ice Nucleation and Its Comparison with Observations. *J. Atmos. Sci.,* 70, 378–409, https://doi.org/10.1175/JAS-D-12-080.1

Pruppacher, H.R. and Klett, J.D. (1997) Microphysics of Clouds and Precipitation. 2nd Edition, Kluwer Academic, Dordrecht, 954 p.

Savre, J., and Ekman, A. M. L. ( 2015), Large‐eddy simulation of three mixed‐phase cloud events during ISDAC: Conditions for persistent heterogeneous ice formation. *J. Geophys. Res. Atmos.,* 120, 7699– 7725. doi: 10.1002/2014JD023006

Young, G., Jones, H. M., Choularton, T. W., Crosier, J., Bower, K. N., Gallagher, M. W., Davies, R. S., Renfrew, I. A., Elvidge, A. D., Darbyshire, E., Marenco, F., Brown, P. R. A., Ricketts, H. M. A., Connolly, P. J., Lloyd, G., Williams, P. I., Allan, J. D., Taylor, J. W., Liu, D., and Flynn, M. J.: Observed microphysical changes in Arctic mixed-phase clouds when transitioning from sea ice to open ocean, Atmos. Chem. Phys., 16, 13945–13967, https://doi.org/10.5194/acp-16-13945-2016, 2016.

---

## Author Comment (AC2) · 14 Dec 2019

**RESPONSE TO REVIEWER 2:**

The manuscript studies the sensitivity of ice particle number concentration on different assumptions about secondary ice production in Arctic mixed phase clouds. … role of secondary ice production and gives directions for further studies. Thus I recommend publication after corrections and additional discussion.

We thank the reviewer for the thoughtful comments that have clearly improved the manuscript. Our responses (black) to each point raised (red) is provided below.

**What is the role of measurement conditions? It is said that the wind is from the West and measurements are performed both over the open water and ice. Droplet concentration seems similar in both, but there is no discussion how ice particle concentration differs and where the presented values are measured.**

An overview of the observed conditions with respect to ice-covered or open-water surface was provided in Jones et al (2018). However since this manuscript remained in the discussion phase, a brief recap on the influence of the surface state on cloud microphysical properties is now added (lines 331-335 in the revised version). In the initial submission, in lines 271-273 we state that we use cloud measurements collected at latitudes higher than $81.7°N$ and within a $9×33$ $km^2$ ice-covered area to evaluate the simulated cloud properties, as ice-covered surface conditions are also prescribed in the LES. Thus all cloud observations shown in Figures 5-7-9 in the initial document are collected over ice.

**Is there any potential for the surface to be the source if ice hydrometeors?**

Blowing snow is associated with strong winds over flat terrain (e.g. Vali et al. 2012; Gossart et al. 2017) or close to mountainous slopes in the vicinity of orographic clouds (e.g. Lloyd et al. 2015; Geerts et al 2015). A general threshold in 2-m wind speed for such events in freshly fallen snow is $7–10$ m s$^{-1}$, with a weak trend toward lower threshold speeds at lower air temperatures (Dery and Yau 1999). Gossart et al. (2017) showed that the height of the blowing snow layer is usually << 500 m, except for stormy cases of heavy mixed events, precipitation and blowing snow ,when it can go up to 1.3 km. In our case, the winds are much weaker, on average ~ 5.8 m s$^{-1}$and the cloud base height is >500 m AGL, while the largest concentrations were observed at ~800 m AGL. A maximum height of 500 m for such phenomena is also recorded in Geerts et al (2017), but required much higher winds than in our case. Thus we believe there is no possibility for blowing snow to impact the examined clouds (this is also mentioned at the end of section 2.2 in the revised text) .

**2) LPM: I have some problems in understanding what the LPM model employed is actually simulating. Does it solve the hydrometeor condensational growth assuming three size classes both for liquid and ice, or is it somehow parameterized how particles grow from some size range to another based on characteristic time parcel is spending in a single updraft. I'm actually surprised that there is no spectral size resolving model employed. Such models should be available and numerically efficient enough to be used in the presented application. In line 139 it is stated "The LPM allows a detailed description of the formation, growth and evolution of cloud droplets and ice particles as they interact with each other". I disagree with this, I would not call three bins detailed what comes to representation of cloud and ice particle size distributions. Thus also coagulation rate and secondary ice production are only approximate, although probably accurate enough to provide first estimates and to be used in this paper.**

The LPM allows all bins to evolve dynamically by predicting their size as a function of

temperature and supersaturation. However the transition from one bin to another is controlled by the timescales. This is based on Yano and Phillips (2011) and is a simple but still convenient framework to parameterize SIP. While more advanced spectral size resolving models would likely offer more accurate predictions of SIP effects, these are computationally expensive and do not allow SIP investigations over a very large parameter space. Here we demonstrate the possibility of using a simplified framework to develop parcel-model based parameterizations for larger scale models. However, we agree that this is not a very detailed model, and we have replaced the word 'detailed' with 'adequate' in the revised text.

**3) MIMICA: Line 145 or later in section 3.1: Maybe you should state explicitly the reason why SIP processes are not directly implemented into MIMICA.**
The original submission had this extensively discussed in the last section. In the revised text, we have moved this discussion to the beginning of section 3.

**4) How does the SIP enhancement work in a case when the ice particle concentration at cloud base in MIMICA is higher than prescribed IN concentration? Does it still enhance the concentration? I assume such conditions to occur frequently in modeled boundary layer cloud.**
At each model time-step and level the LES estimates the number of new nucleated particles = INP (using DeMott in the revised version) - existing $N_{ice}$. If this is negative, nucleation of new particles is assumed to not occur. This treatment can be found in widely used microphysics schemes (e.g. Morrison et al. 2005 in WRF). In the case that SIP is activated, the only difference is that: new nucleated particles = INP $\times$ $SIP_{factor}$ - existing $N_{ice}$ (thus if the outcome is negative, SIP does not enhance concentrations anymore). This methodology is better explained now in section 3.1 of the revised manuscript to avoid confusion.

**5) Line 283: "The mean observed INP concentration is 0.006 L-1 and never exceeds 0.05 L-1". From where does these numbers come from? The conditions are really warm for heterogeneous ice nucleation, with modelled values at minimum -6.5 degrees and measured even warmer. What aerosol particles are active in such a warm temperature.**
These statistics are based on all the flight data collected on 23 July 2013, between 10-11 UTC, at various latitudes, longtitudes and altitudes. They aim to provide a more general overview and also allow for comparison with other ACCACIA cases (e.g. Lloyd et al. 2015), in which INP conditions are estimated and presented in a similar way. However, we acknowledge that this vague discussion is confusing to the readers as no reference to the thermodynamic conditions is made. This is now corrected in the revised manuscript. 0.006 $L^{-1}$ is the mean concentration is for the whole flight, which sampled at temperatures between $\sim$ -10 $^{o}C$ – 0 $^{o}C$ and specific humidity $\sim$ 2.5–5 g $m^{-3}$. The maximum INP concentration is observed at $\sim$ T= -10 $^{o}C$ and Qv=2.5 g $m^{-3}$. However the maximum ICNC occurs at T$\sim$-5$^{o}C$, much warmer conditions than those that maximum INPs are measured. Hence, measurements strongly indicate the occurrence of SIP.

Demott aerosol-aware parameterization predicts INP=0.03 $L^{-1}$ around -6.5$^{o}C$ (the coldest simulated temperatures), which interestingly is in very good agreement with the upper limit of INPs measured in the Arctic (Figure 7 in Wex et al. 2019). We don't know the chemical composition of ice nuclei at these temperatures, however Wex et al. measured all types of aerosols, including bioaerosols.

**6) Within MIMICA it would be possible to track temperature dependent IN concentration. How would this more realistic approach change the simulations? In comparison to observations it would have been interesting to see if the spread in modelled data is as wide as in observations. When I look at modelled data, I am really surprised how small standard deviation there is in the output. Enhancement should depend quite strongly on the updraft at the cloud base based on Figures 4, 6 and 8.**

The DeMott aerosol-aware temperature-dependent parameterization is now implemented in the LES, with mean observed aerosol concentrations as input. However, the original scheme predicts INP=0.03 $L^{-1}$ around -6.5°C: even if this very low INP concentration is likely realistic (Figure 7 in Wex et al. 2019), when prescribed in the LES simulations, it does not produce any ice. For this reason we have to consider the uncertainty in DeMott parameterization which is a factor of 10. Multiplying DeMott×10 yields very large INP concentrations (~0.3 $L^{-1}$) near cloud top (minimum -6.5°C), which is unrealistic for warm subzero temperatures (Figure 7 in Wex et al. 2019). For this reason, in our CNTRL simulation we apply DeMott×5. The original DeMott scheme, DeMott×10 and DeMott×100 are presented as sensitivity tests.

Below we show the mean profiles produced by the LES after spin-up period (no SIP is activated): DeMott×5 predicts concentrations ($N_{ice}$) varying from 0.007 $L^{-1}$ at cloud base (~ -3°C) to 0.11 $L^{-1}$ near cloud top. An interesting finding is that $N_{ice}$ profile does not match the vertical distribution of INPs predicted by the ice nucleation scheme (Figure 1). This is likely due to the effect of cloud mixing of ice crystals, as this more homogeneous profile develops within the first hour of simulation. If SIP was directly implemented in the LES, $N_{ice}$ profiles would be used to calculate ice-ice collisions. Hence we use the $N_{ice}$ profiles as input to the LPM: a mean INP concentration of 0.007 $L^{-1}$ is prescribed at cloud base, while as the parcel ascends the new nucleated crystals estimated with a nucleation rate based on DeMott×5 (the product of updraft velocity, an assumed lapse rate of 6 K $km^{-1}$, and the temperature derivative of the INP estimates) until a maximum value of 1.1 $L^{-1}$ INPs is reached; this is the maximum $N_{ice}$/INP concentration produced by the LES near cloud top (Figure 1). This discussion and the Figure below have been as added to the revised Supporting Information.

[Figure]

**Figure 1**: Mean LES profiles of INP and $N_{ice}$ concentrations after spin-up period.

Furthermore, we would like to clarify that the small standard deviation in the LES simulations is not due to the prescribed INP conditions. The LES reaches a quasi-steady state after a few hours and the presented statistics are derived from that period of equilibrium. LPM is run over a relatively large parameter space, but the LES conditions are not that variable, especially in the quasi-steady state. In Figure 2a below it is obvious that only the low updraft conditions are representative of the simulated cloud (stronger updrafts are basically outliers with relative frequency < 0.2%). Also we cover in-cloud temperature conditions with the LPM; however, the parameterization in the LES is eventually a function of cloud base temperature, whose range is much narrower (Figure 2b). Since running the LPM is computational cheap, testing a larger parameter space was no problem for us: it ensures that all possible conditions are accounted for. However, now we understand that the wider thermodynamic conditions presented in Figures 4,6,8 in the previous manuscript might confuse the reader. For this reason we have moved the LPM look-up tables to the Supporting Information, while only the dominant conditions are discussed in the main text.

[Figure]

**Figure 3**: Relative frequency distribution for hourly outputs of 3D LES fields of (a) incloud updraft velocities ($u_z > 0.05$ m s$^{-1}$) and (b) in-cloud (red) and cloud-base (blue) temperatures. Dashed lines represent mean values.

**7) Line 463: "A main challenge in parameterizing BR is that a correct spectral representation of the ice crystals is required, which is more feasible in bin microphysics schemes". This is true, and the same limitations holds for all cases when temperature dependent ice nucleation or secondary ice production is included. If the number concentration is tuned to be correct, the size distribution and total mass is probably wrong due to given shape for size distribution.**
This is a common problem in bulk microphysics schemes. The very detailed bin microphysics schemes however are computationally expensive and that's why they are not widely used in weather prediction and climate models. For this reason we suggest that LPMs can serve as a efficient way to parameterize SIP in large-scale models: although several simplifications are still employed, they likely can offer a reasonable estimate of the order of magnitude of SIP multiplication in different cloud states.

**8) Jones et al., 2018 is not accepted for publication, so it should not be cited.**
We have removed this citation in the revised manuscript

**9) Schwarzenboeck et al., 2009 title is "Indications for stellar-crystal fragmentation in Arctic clouds"**
Thank you, corrected

**References:**

Déry, S. J. and Yau, M. K.,1999: 'A Climatology of Adverse Winter-Type Weather Events', J. Geophys. Res. 104(D14), 16,657–16,672.

Morrison, H., Curry, J.A., & Khvorostyanov, V.I. (2005). A New Double-Moment Microphysics Parameterization for Application in Cloud and Climate Models. Part I: Description, . *Atmos. Sci.,* 62*,* 3683-3704 62

Vali, G. , Leon, D. and Snider, J. R. (2012), Ground-layer snow clouds. Q.J.R. Meteorol. Soc., 138: 1507-1525. doi:10.1002/qj.1882

Geerts, B., B. Pokharel, and D.A. Kristovich, 2015: Blowing Snow as a Natural Glaciogenic Cloud Seeding Mechanism. *Mon. Wea. Rev.,* 143, 5017–5033,https://doi.org/10.1175/MWR-D-15-0241.1

Lloyd, G., Choularton, T. W., Bower, K. N., Gallagher, M. W., Connolly, P. J., Flynn, M., Farrington, R., Crosier, J., Schlenczek, O., Fugal, J., and Henneberger, J.: The origins of ice crystals measured in mixed-phase clouds at the high-alpine site Jungfraujoch, Atmos. Chem. Phys., 15, 12953–12969, https://doi.org/10.5194/acp-15-12953-2015, 2015.

Lloyd, G., Choularton, T. W., Bower, K. N., Crosier, J., Jones, H., Dorsey, J. R., Gallagher, M. W., Connolly, P., Kirchgaessner, A. C. R., and Lachlan-Cope, T.: Observations and comparisons of cloud microphysical properties in spring and summertime Arctic stratocumulus clouds during the ACCACIA campaign, Atmos. Chem. Phys., 15, 3719-3737, https://doi.org/10.5194/acp-15-3719-2015, 2015.

Gossart, A., Souverijns, N., Gorodetskaya, I. V., Lhermitte, S., Lenaerts, J. T. M., Schween, J. H., Mangold, A., Laffineur, Q., and van Lipzig, N. P. M.: Blowing snow detection from ground-based ceilometers: application to East Antarctica, The Cryosphere, 11, 2755–2772, https://doi.org/10.5194/tc-11-2755-2017, 2017.